



# A Lightweight Vortex Particle-Mesh Library for Variable-Fidelity Simulation of Wind Turbine Wakes

Joseph Saverin[1]

Chair of Fluid Dynamics, Hermann Föttinger Institute, Technische Universität Berlin, Berlin, Germany

**Correspondence:** Joseph Saverin (j.saverin@tu-berlin.de)

**Abstract.** In this work, the implementation and validation of a lightweight vortex-particle mesh solver is described. Within this method, the flow field is discretised with vorticity-carrying Lagrangian particles. Field quantities are calculated on a homogeneous rectangular grid and mapped to the instantaneous particle positions. A fast Poisson solver is employed to efficiently calculate the velocity field on the grid using fast Fourier transforms. The vorticity field is updated by applying the vorticity transport equation, the terms of which are extracted from the flow field using finite differences. The flow solver is validated by simulating an unsteady vortex ring. A nested grid region is generated which allows for the flow field representation of lift-generating bodies on the main grid. This secondary grid approach allows for the use of arbitrary lifting body models. Here, a nonlinear lifting line method has been applied and validated against an analytical solutions for an elliptical wing. The solver is validated for helical wake configurations through comparison with an idealized Betz rotor. Finally, the application to the simulation of wind turbine wakes is demonstrated through comparison with experimental results of rotor loads and wake hot-wire measurements performed in the Mexnext wind tunnel measurement campaign.

## 1 Introduction

The opportunity to reduce restrictions on turbine spacing in a wind farm can lead to a range of material savings as well as advantages for both operation and maintenance of onshore and offshore installations. These advantages can reduce capital and operational costs, making wind energy an even more competitive option for the generation of renewable energy or the production of green hydrogen. A driving factor in the selection of turbine spacing is wake interaction, whereby the wake of one or more turbines impinges on a downstream turbine, increasing unsteady blade and rotor loads and fatigue of turbine components (Lee et al., 2012). In addition to the interaction between turbines, the wake also interacts with the terrain, turbulent inflow and the atmospheric boundary layer. These factors combined result in a highly complicated and turbulent velocity field which is only partially amenable to simplified modelling. One suggested methodology for the reduction of the turbine spacing is to actively actuate the upstream turbine in order to amplify the wake breakdown and recovery processes either through their interaction with ambient turbulence, the atmospheric shear layer, or through inherent hydrodynamic instabilities within the wake topology (Frederik et al., 2020; Cheung et al., 2024). An alternative approach is to yaw the rotor in order to deflect the turbine wake away from downstream rotors (Howland et al., 2019). A comprehensive overview of the suggested concepts is given in Korb et al. (2023). Whether the purpose is to improve understanding of the inherent physical processes occurring



in wake breakdown and recovery, quantify turbine aerodynamic interactions, optimise turbine components, or design novel control strategies for the aforementioned approaches, there is a strong need for efficient and easily parameterised models for simulating the wake of wind turbines. A range of approaches exist for the simulation of wind turbine wakes for both single turbines and full farm applications. A brief overview is given here of the state of the art and exemplary software tools, in order
to provide a context and domain of applicability for the implemented tool.

For highly-resolved simulations of the atmospheric boundary layer and turbulent inflow field the family of large eddy simulation (LES) tools allow for accurate unsteady treatment of the flow field by using a filtered form of the Navier-Stokes (NS) equations and modelling the effects of small turbulent scales with a sub-grid scale model (Martínez-Tossas et al., 2015). These
methods are usually based on either the finite-volume or finite-difference methods for resolution of grid quantities. For users with access to super-computing devices, the tools developed by the ExaWind organisation of the national renewable energy laboratory (NREL) NaluWind (Sprague et al., 2020) and AMRWind (Ananthan et al., 2020) are applicable for high fidelity simulations which enable detailed investigation of flow field physics. An alternative, more generally applicable alternative is the SOWFA library (Churchfield and Lee, 2012) based on the open-source solver OpenFOAM (Weller et al., 1998). This can
be directly coupled to NREL's holistic aero-hydro-servo-elastic turbine plant simulator OpenFAST (Jonkman, 2013). An alternative approach to the use of finite volume codes is the use of the spectral-element method. Here, the solution on the grid is expanded in terms of local basis functions (Kleusberg et al., 2017). This was applied successfully to investigate the impact of turbine motion on the stability of a wind turbine wake in offshore scenarios in Kleine et al. (2021).

An alternative approach to the numerical treatment of a wind turbine wake which has gained significant traction in recent years is the Lattice Boltzmann Method (LBM). In this method, rather than discretising the NS equations, flow quantities on a regular cartesian grid are updated through the application of streaming and collision processes. Although this approach is rooted in the treatment of molecular gas dynamics, it can be applied successfully to fluid flows (Banari et al., 2015). This method is highly amenable to treatment with parallel architectures due to locality of the grid updating processes. LBM was
successfully applied to simulate active wake excitation with the helix technique by Korb et al. (2023). A reinforcement learning approach was used to tune the control of a multi-turbine array while applying the helix technique by Korb et al. (2021).

The final class of methods can broadly be categorised as vortex methods, whereby the vorticity field of the flow is discretised. A range of approaches exist depending on the desired level of accuracy and field equation to be solved. Assuming a
quasi-inviscid velocity field, the vortex filament method may be applied. In this method, vortex tubes are discretised as 3D line elements and the induced velocity calculated with the Biot-Savart Law (Phillips and Snyder, 2000). This method has been practically applied to helicopter wake dynamics (Leishman et al., 2002) or to wind turbine wake aerodynamics (Grasso et al., 2011). The vortex filament method may additionally be applied to derive analytical results as was done for the problem of helical wake stability by Widnall (1972). Rather than discretising the field with line elements, a higher degree of connectivity can
be achieved through the use of the vortex particle (VP) method (Winckelmans and Leonard, 1993). In this case, the vorticity





transport equation is applied to calculate the evolution of the flow field. This approach allows for the treatment of shear stress and vortex filament dissipation and reconnection. In addition, models fur turbulent shear stress may be included as was done by Winckelmans (1995). Direct evaluation of the velocity field using the VP method scales as $\mathcal{O}(N^2)$ and is therewith untenable for large number of particles. This complexity may be reduced to $\mathcal{O}(N \log N)$ or $\mathcal{O}(N)$ through the use of tree-based methods

such as the fast multipole method (FMM) (Greengard and Rokhlin, 1997) or the comparable multilevel multi-integration cluster (MLMIC) method (van Garrel et al., 2017; Saverin et al., 2018). An alternative approach to reducing computational complexity is to resolve field quantities on an Eulerian grid in order to update the position and strength of Lagrangian vortex markers as pioneered in the work of Chatelain and Koumoutsakos (2010). This approach is referred to as the vortex particle-mesh (VPM) method. The use of a regular grid allows for the application of efficient fast Fourier transform (FFT) algorithms for the solution

of the Poisson equation for the stream function or velocity field on the grid. The VPM method has been successfully applied to numerous problems including the modelling of horizontal and vertical axis wind turbines by Chatelain et al. (2013) and Chatelain et al. (2016), respectively. The application to immersed lifting line problems was also investigated in Caprace et al. (2020b). The VP method is naturally amenable to the development of hybrid filament-particle methods as filaments are easily represented within a particle framework. This feature is exploited in the work carried out here.


Within this paper the implementation and validation of a VPM solver is described. The solver integrates the vorticity transport equation on a regular grid within a rectangular three-dimensional domain and is conceptually similar to the solver described in Chatelain et al. (2013). A novel approach to the treatment of the lifting body has been implemented through the use of a second, embedded grid region which has compact support of the lifting body representation. Within this embedded grid, a non-

linear lifting line approach is adopted which models the lifting surface with vortex filaments (Van Garrel, 2003). An iterative procedure is applied to calculate the unsteady circulation distribution along the lifting line. The solver is wrapped within a modular `Wake` class, which may be interchanged in order to facilitate comparison with different rotor and wake models. In Section 2 the implementation and features of the solver are described. The validation of the model is carried out in Sect. 3. Three scenarios are simulated. i) The evolution of a translating vortex ring is simulated in Sect. 3.1. ii) The loads and wake of

an ideal elliptical wing is simulated in Sect. 3.2. iii) The model is applied to the simulation of both an ideal propeller and an experimental wind turbine rotor in Sect. 3.3.

## 2 Implementation and Features

Within the solver, the VPM method is applied to simulate the evolution of an incompressible, single-phase fluid flow (Chatelain et al., 2013). In this approach, the field is described through Lagrangian vorticity markers or particles. The position and vorticity

of these particles are updated according to the vorticity transport equation - Eq. (3). Field quantities are resolved on a rectilinear grid with uniform mesh spacing $h_x$, $h_y$ and $h_z$, leading to cells of volume $dV = h_x h_y h_z$. The entire domain $\mathcal{D}$ is therefore represented as a rectangular prism with lengths $l_x$, $l_y$ and $l_z$, giving a total volume $V = l_x l_y l_z$. $\mathcal{D}$ is assumed to have the compact support of the vorticity field $\omega$. Field quantities are resolved either on cell vertex (*regular*) or at cell centre (*staggered*)



positions. The grid has $n_x$, $n_y$ and $n_z$ cells for a total of $N_t = (n_x + 1)(n_y + 1)(n_z + 1)$ nodes for the regular configuration
and $N_t = n_x n_y n_z$ nodes for the staggered configuration.

## 2.1 Grid Objects and Mapping Procedures

Grid variables are stored as array objects in a *structure of arrays* format in order to optimise memory accesses. Each array
object is stored as follows:

$$\bar{\bar{A}} = \begin{bmatrix} d_x, & d_y, & d_z, & \omega_x, & \omega_y, & \omega_z \end{bmatrix}, \tag{1}$$

where $d_i$ and $\omega_i$ are arrays that contain the components of particle displacement $d_i$ and vorticity $\omega_i$, respectively. Each array
has $N_t$ elements, corresponding to each solution grid node. Two grid objects are utilised in the solution procedure: the Eulerian
grid $\mathcal{G}_E$ and the Lagrangian grid $\mathcal{G}_L$. The Eulerian grid stores field properties at the stationary Eulerian grid nodes. On $\mathcal{G}_E$
the displacement arrays $d_i$ store the node absolute spatial position. These are calculated once at initialisation of the solver
and remain unchanged. The absolute node positions are utilised for specification of initial vorticity distributions and in the
calculation of flow field diagnostics. The Lagrangian grid contains field properties at the spatial position of the Lagrangian
particles. The grid is instantiated in such a way that the particles coincide with the Eulerian grid nodes, such that the relative
displacement vectors are initially zero $d_i = \mathbf{0}$. As the particle set is integrated in time, the Lagrangian particles are advected
and the displacement vectors $d_i$ represent the absolute displacement relative to the corresponding Eulerian grid node. This is
illustrated for a simpified 1D case in Fig. 1.

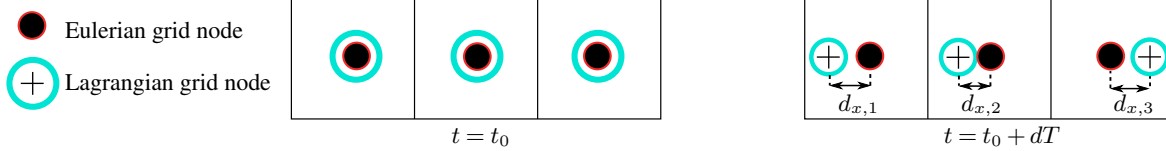

**Figure 1.** The relation between Eulerian and Lagrangian grid nodes illustrated for simplicity on a 1D grid with a single row of cells. The
timestep $t_0$ represents either initialization of the Lagrangian grid or the positions after a redistribution step. After a timestep of size $dT$ has
been taken, the Lagrangian node displacements are stored in arrays $\mathcal{G}_{L(1-3,i)}$.

Key to the VPM method is that flow field quantities required for evolution of the vorticity field $\boldsymbol{\omega}$ are calculated exclusively
on the Eulerian grid, allowing for the use of algorithms which exploit the regularity and symmetries of the grid. Field quantities
are mapped between the Eulerian and Lagrangian grids by using moment-conserving mapping functions. A range of such
mapping functions exist including $M_2$, $M_4'$ and $M_6^*$ (Cottet and Koumoutsakos, 2000; Bergdorf, 2006). Higher-order mappings
ensure that higher moments of the field are conserved. These mappings are carried out in 3D in a tensorial fashion such that
the mapping factor $f$ is calculated as the product of the appropriate $f_i$ in each spatial direction: $f = f_x f_y f_z$. This process is
demonstrated graphically in Fig. 2 for the mapping of a single particle on $\mathcal{G}_L$ to the corresponding neighboring particles on
$\mathcal{G}_E$. The mapping procedure occurs as follows: For each particle, the particle displacements stored on the $\mathcal{G}_L$ grid are used to
specify the factors $f_{x,i}$, $f_{y,i}$ and $f_{z,i}$ $(i = 1, 2, \ldots, n_s)$, where $n_s$ is the width of the mapping stencil. These are stored on a map



coefficient array $\mathcal{M}_{ij}$ which has dimension $(N_t, dN_s)$ where $d$ is the dimension of the problem. In the second step, for each grid node, the $\mathcal{M}_{ij}$ array is used to sum the contributions of each surrounding $\mathcal{G}_L$ node to each $\mathcal{G}_E$ grid node. This procedure ensures that mapping coefficients are calculated and stored only once for each mapping and avoids data races in parrallelisation. Additionally, after the calculation of flow quantities on the Eulerian grid, $\mathcal{M}$ may be applied to map the Eulerian grid quantities back to the Lagrangian particles $\mathcal{G}_E \rightarrow \mathcal{G}_L$.

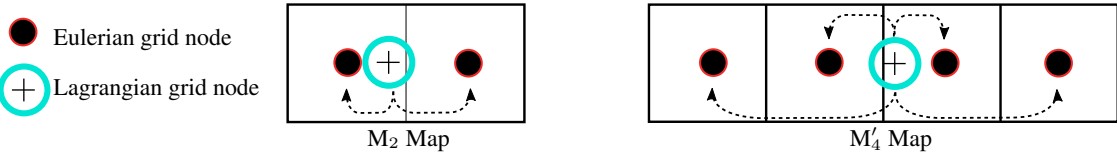

**Figure 2.** The mapping between a Lagrangian grid node and an Eulerian grid node illustrated for simplicity on a 1D grid with a single row of cells. Each dashed line represents a mapping factor which represents the contribution of the Lagrangain node to the surrounding Eulerian nodes as a function of the displacement distance. The $M_2$ mapping has a stencil of width $n_S = 2$, the $M_4'$ mapping have a stencil width of $n_S = 4$.

## 2.2 Particle Position Update- Poisson Equation

At each time step, the particle positions on the $\mathcal{G}_L$ grid are updated by advecting the particles with the local velocity field. This is calculated on the Eulerian grid through the application of a fast Poisson solver for the velocity field:

$$\nabla^2 \boldsymbol{u} = -\nabla \times \boldsymbol{\omega} \,, \tag{2}$$

where $\boldsymbol{u}$ is the velocity of the field. The open-source fast Poisson library `SailFFish` has been applied to resolve the velocity field on the grid (Saverin, 2023). The `SailFFish` library makes use of the fast Fourier transform to solve the Poisson equation in spectral space. The curl operator in Eq. (2) has been applied directly in spectral space through the use of spectral differentiation. Free-space boundary conditions have been applied using Hockney's domain-doubling technique to treat the unbounded problem as a periodic problem on an extended domain (Hockney; Eastwood and Brownrigg, 1979). This greatly increases the overhead as the domain must be doubled in each direction. Alternative formulations may be applied to reduce memory footprint and calculation overhead including sequential transforms in each spatial direction as is done in the `FLUPS` library of Caprace et al. (2021). An alternative approach to the solution is achieved through a restatement of the unbounded problem into a mixed dirichlet problem as was done in James (1977). A variety of regularisations of the Green's solution to the Poisson equation are available for use for the convolution step within `SailFFish` depending on the desired convergence behaviour and application of the solution. These include Gaussian regularisations of order $2, 4, \ldots, 10$ or the pseudo-spectral Green's function derived by Hejlesen et al. (2019). The regularisation should be chosen such that the convergence behaviour aligns with that of the finite difference schemes for calculating the field quantities in order to ensure consistency.



### 2.3 Particle Strength Update- Vorticity-Transport Equation

The update of the particle strength occurs through the resolution of the vorticity transport equation:

$$\frac{d\boldsymbol{\omega}}{dt} = (\boldsymbol{\omega} \cdot \nabla)\boldsymbol{u} + \nu\nabla^2\boldsymbol{\omega} + \nabla \cdot T^M \,, \tag{3}$$

where $\nu$ is the kinematic viscosity of the fluid and $T^M$ is the sub-grid scale (SGS) stress tensor. The gradients terms $\nabla\boldsymbol{u}$ and
$\nabla^2\boldsymbol{\omega}$ are calculated on the Eulerian grid using central **f**inite **d**ifferences (FD) of variable order. Due to the symmetric nature of the node alignment, the FD relative node indices are calculated at initialisation of the solver and stored in a FD template. The central FD schemes are naturally not applicable at the boundary of the domain. For this reason FD exclusion zones are specified at solver initialisation. As FDs are not calculated in boundary regions, this implies that during the simulation the vorticity must vanish at the boundary of the domain $\mathcal{D}$. Extensions to allow for nonzero velocities at the domain boundaries is planned for future work. This would allow, for example, the treatment of a turbulent inflow field. A range of models exist for the SGS tensor (Cocle, 2007), however for the results shown in this work, $T^M = 0$.

### 2.4 Time Marching & Numerical Stability

Flow quantities resolved are mapped $\mathcal{G}_E \rightarrow \mathcal{G}_L$ using the same mapping coefficients used to carry out the mapping $\mathcal{G}_L \rightarrow \mathcal{G}_E$. This results in a rate of change vector corresponding to the Lagrangian grid values which allows for the straightforward implementation of a variety of integration schemes. Currently within the VPM solver the following explicit integration schemes are implemented: Eulerian forward, explicit midpoint, Runge-Kutta 2nd, 3rd and 4th order and a combined Adams-Bashforth ($\omega_i$) / Leapfrog ($d_i$) second order scheme as described in Cocle et al. (2008).

#### 2.4.1 Particle set remeshing

Previous studies have demonstrated that strong distortions of the Lagrangian grid give rise to overly dense or empty regions of particles which can lead to divergence of gradient terms (Koumoutsakos, 1997). A method to overcome this is to routinely remesh the particle grid onto a regular lattice. This is effectively equivalent to the initial step in the rate of change calculation, whereby the vorticity distribution on the Lagrangian grid is mapped to the Eulerian grid. In the case of a remeshing step, the Lagrangian particles strengths are simply copied from the Eulerian mesh and the relative displacement for all particles is reset to zero. This has implications for time-stepping routines. For integration schemes requiring time derivatives from previous timesteps (e.g. Adams-Bashforth), an equivalently high-order multistep procedure (such as a Runge-Kutta scheme) must be executed directly after remeshing in order to ensure consistency. The remeshing schemes employ the mapping functions described in Sect. 2.1. These mapping functions require a relative displacement $d/h \leq 1.0$, which leads to a constraint on the allowable particle relative displacement before remeshing must be applied. This limit constrains the **C**ourant-**F**riedrichs-**L**ewy number $\text{CFL} = |\boldsymbol{u}|_{max}\Delta t/h$ to be below unity. The remapping procedure may be reformulated to avoid this constraint, this however will violate the kernel abstraction principle currently applied in the solver and is left to future work. It should be noted





that the remeshing procedure is equivalent to the application of a hyperviscosity operator on the flow field (Koumoutsakos, 1997; Cocle, 2007).

### 2.4.2 Magnitude Filtering

As a result of the finite width of the remeshing stencil, the domain of the vorticity expands during each remeshing step. This
has the effect of expanding the support of the vorticity which necessitates gradually expanding the boundary of $\mathcal{D}$. A method to limit this is to remove low-energy regions of the flow by carrying out a magnitude filtering. During evaluation of the particle set diagnostics, the maximum value of vorticity $|\boldsymbol{\omega}|_{max}$ is extracted. During magnitude filtering, all particles which have a vorticity magnitude $|\boldsymbol{\omega}| < f_{mag}|\boldsymbol{\omega}|_{max}$ are removed where $f_{mag}$ is the magnitude filtering factor. The total vorticity removed is appended to the remaining nonzero magnitude particles, to ensure consistency in the first-order particle set diagnostics.

### 2.4.3 Particle set reprojection

It can be easily shown through kinematic considerations that the vorticity field must remain divergence-free for all time: $\nabla \cdot \boldsymbol{\omega} = \mathbf{0}$. This is not guaranteed in a time-marching simulation by the formulation of the VPM expressed through Eqs. (2) and (3). A novel method proposed by Cottet and Poncet (2003) is to re-project the vorticity field onto a divergence-free basis. This is done by calculating the divergence of the vorticity field on the grid using FD. A correction field $F$ is then calculated as
$\nabla^2 F = \nabla \cdot \boldsymbol{\omega}$. Upon calculation of $F$, the vorticity field can be updated with $\boldsymbol{\omega}_{new} = \boldsymbol{\omega}_{old} - \nabla F$. Unlike the unbounded solver applied for the solution to Eq. 2, $F$ must also have compact support, requiring a homogeneous Dirichlet boundary condition. This solver type is available within `SailFFish` and is exploited for this correction. This type of solver is in fact much more efficient than the unbounded solver, as the domain volume need not be extended as required by the Hockney domain-doubling technique. In addition, a Dirichlet fast Poisson solver makes use of efficient real-to-real (R2R) discrete sine transforms. Particle
reprojection is carried out following a remeshing step, such that the relative particle displacements are zero. This ensures that the reprojection process can be carried out entirely on the Eulerian grid.

### 2.5 Treatment of Lifting Bodies

As the lifting body generally occupies a relatively small portion of the entire flow domain, an embedded sub-domain $\mathcal{D}_\phi \in \mathcal{D}$ is generated for the storage and mapping of lifting body elements as visualised in Fig. 3. Generating a second, embedded grid
within the flow domain has numerous potential advantages for the solution procedure chosen. For high resolution or blade-resolved treatments of the flow around the lifting body, alternate solvers including traditional finite volume solvers may be employed and the solution within this domain mapped to the bounding domain $\mathcal{D}$. A similar approach to this was taken in the work by Spyropoulos et al. (2022). Two methods of solution within the embedded domain $\mathcal{D}_\phi$ are described in the following.





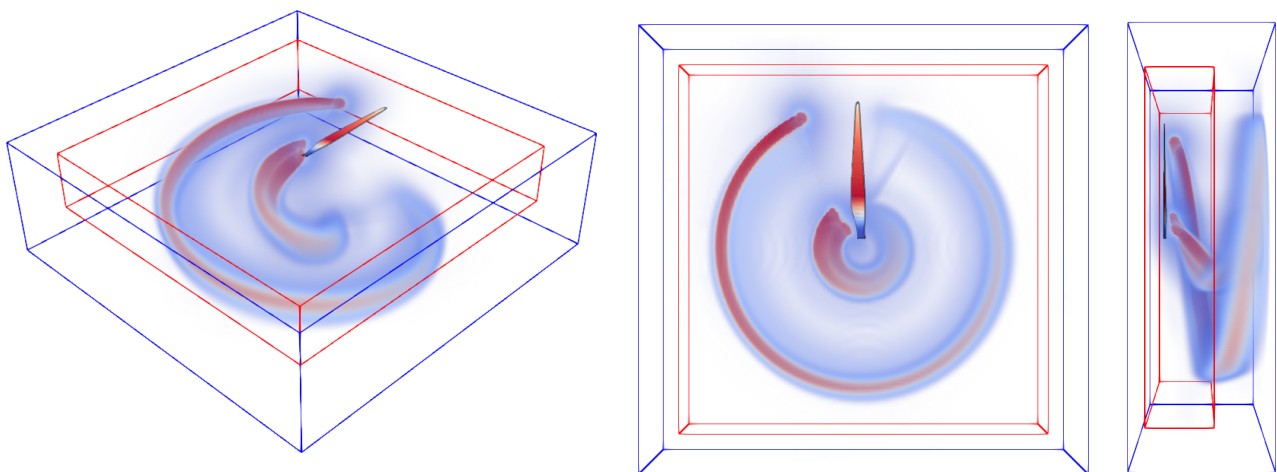

**Figure 3.** The subdomain $\mathcal{D}_\phi$ (red boundary) is contained within the global domain $\mathcal{D}$ (blue boundary). A volume rendering of the wake of a single rotation of a 1-bladed rotor is shown here for visualisation purposes.

### 2.5.1 Lifting Line Method

A suitable method for the treatment of wing or rotor aerodynamics is the lifting line method developed by Prandtl (1918). In this method, the lifting body and wake is discretised with vortex filaments in the chordwise and spanwise directions. This method has been advanced by numerous authors to account for realistic planforms and airfoil sections by using sectional airfoil lift, drag and moment coefficients (Phillips and Snyder, 2000; van Garrel, 2003). This approach is hereafter referred to as the **n**onlinear **l**ifting **l**ine (NLL) method. The circulation distribution along the lifting body requires calculation of the total incident

velocity $u_{tot}$ at a set of control points distributed along on the span of the lifting body. $u_{tot}$ is calculated through a superposition of the following velocity field components:

$$u_{tot} = u_\infty - u_{mot} + u_\phi + u_\omega \ , \tag{4}$$

where $u_\infty$ is the irrotational freestream, $u_{mot}$ is the incident velocity due to the motion of the lifting body, $u_\phi$ is the velocity induced by the lifting body and near-wake filaments and $u_\omega$ is the velocity field mapped from the Eulerian grid. The com-

ponent $u_\phi$ is calculated with a filament-style approach, whereby the $\delta$-factor regularisation as described in van Garrel (2003) has been applied. At each timestep, the components of $u_{tot}$ allow for the calculation of the sectional angle of attack. These, together with the tabulated airfoil lift coefficients, can be used to carry out a fixed-point iteration to determine the circulation distribution along the lifting line. The same equation is used to advect the wake nodes in the near-field of the lifting body. The NLL method has been implemented in a separate aerodynamics module coupled to the VPM solver and is validated in Sect. 3.2.


   The filament representation of the lifting body and near-wake is converted at each time step to an equivalent particle representation $\omega_\phi$ on the embedded grid $\mathcal{D}_\phi$. Once a wake filament reaches a specified age in timesteps, $N_{conv}$, that filament is



removed from the near-field filament set and its particle representation is mapped to the Eulerian grid $\mathcal{G}_L$- see Fig. 4 b). This process represents the generation of the vorticity profile on the flow field by the lifting body. In order to treat the effect of the

lifting body on the flow field, the $\omega_\phi$ field on the embedded grid is superimposed on the global grid $\mathcal{D}$ for the calculation of the velocity field $u$. This vorticity component is however not included in the calculation of shear stress on the grid (Eq. 3). This effectively decouples the generation of vorticity at the lifting body from the main flow field and avoids issues caused by insufficiently resolving the flow field around the lifting body. This implies that different velocity field regularisations are being applied in the embedded and bounding domains $\mathcal{D}_\phi$ and $\mathcal{D}$, respectively. Under circumstances where gradients are significant

in this region, this may lead to discontinuities at the interface and care must be exercised. For the work carried out here no difficulties were observed.

|  (a)  |  (b)  |  (c)  |
|---|---|---|
| 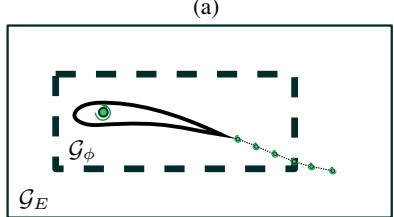 | 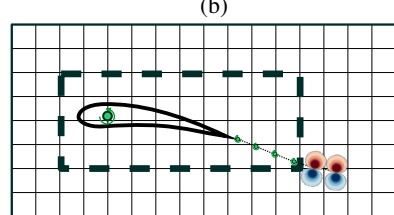 | 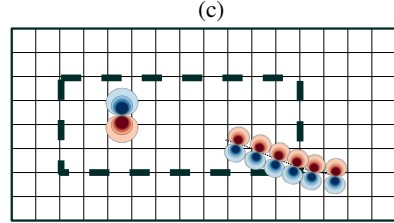 |

**Figure 4.** Visualisation of zoning approach for calculation of velocity fields incident upon lifting bodies. (a) Eulerian grid, embedded grid and near wake. (b) Velocity field for calculation of near-wake and lifting body. (c) Velocity field of Eulerian grid including superposed influence of lifting body vorticity $\omega_\phi$.

### 2.5.2   Actuator Line Model

An alternative approach which avoids the resolution of the lifting body is the **a**ctuator **l**ine (AL) method introduced by Sørensen and Shen (2002) for finite volume simulations. As with the NLL approach, in the AL method the aerodynamic influence of the

blade is modelled through a lower-dimensional representation of the lifting line. The velocity at a set of spanwise positions is sampled to determine the local sectional angle of attack. Whereas in the NLL model the sectional angle of attack is calculated at a single coordinate at each spanwise evaluation point, in the AL model a line or surface average is found by sampling points around the given evaluation point. This angle of attack is again used together with tabulated lift and drag coefficients to quantify the sectional loading. A key difference between the NLL and AL models is how the forcing is applied to the flow field. In the AL

method, forcing is introduced as an additional impulse term into the NS equation which is mollified through the application of a regularisation kernel. A range of 1D, 2D and 3D mollifications may be applied allowing for the introduction of a regularising length scale at different blade positions (Churchfield et al., 2017; Schollenberger et al., 2020). The VPM equivalent of this approach is the immersed lifting line (ILL) method, whereby the vorticity field of the blade model is mollified (Caprace et al., 2018). The ILL method allows for a blade treatment comparable to the actuator line method (Martinez-Tossas et al., 2017) and

the incorporation of higher-order effects such as airfoil parasitic drag (Caprace et al., 2020a). This approach has not yet been implemented into the solver and is considered as future work.





## 3 Validations

A range of cases are simulated here to demonstrate the validity of the VPM solver. An unbounded, unforced vorticity field is first investigated in Sect. 3.1. An elliptical wing section submerged in a constant inflow is simulated in Sect. 3.2 to demonstrate the application of the solver to a lifting body. The solver is then applied to the case of a rotating turbine in Sect. 3.3. Comparison will be made to both analytical results and experimental results. Each section is structured as follows: First a steady case is simulated, for which the solution does not require timestepping. Following this, an unsteady case is simulated to demonstrate the accurate time evolution of the vorticity and velocity fields.

### 3.1 Unbounded Vortical Flow: Viscous Vortex Rings

A suitable test case to demonstrate the evolution of the vorticity field is a thin viscous vortex ring. The availability of analytical results for this case allow for a direct evaluation of the velocity field along with the early evolution of the flow field. Initially the kinematics of the ring and analytical solutions will be described. The steady case of the initial velocity field induced by a vortex ring will then be investigated. Following this, the unsteady case of a translating vortex ring will be simulated.

#### 3.1.1 Kinematic Description

The vortex ring has radius $R$, coresize $\delta$ and integral circulation $\Gamma$. The initial vorticity field is specified by the following axysymmetric vorticity distribution:

$$\omega_\theta(r,x) = -\frac{\Gamma}{\pi\delta^2}\exp\left(-\frac{x^2+(r-R)^2}{\delta^2}\right)\,, \qquad \omega_x(r,x) = 0\,, \tag{5}$$

where $r^2 = y^2 + z^2$ and $\theta = \tan^{-1}(z/y)$. The behaviour of the ring can be characterised by the rotational Reynolds number $\mathrm{Re} = \Gamma/\nu$, the characteristic time $t_f = R^2/\Gamma$ and the nondimensional coresize $\epsilon = \delta/R$. As a result of the self-induced velocity field, the ring translates in the positive $x$-direction. The first analytical result for the translational velocity of the viscous vortex ring was given by Saffman (1970) as:

$$\mathcal{U}_S = \frac{d\mathcal{X}}{dt} = \frac{\Gamma_0}{4\pi R}\left[\log\frac{8}{\epsilon} - \beta_0 + \mathcal{O}(\epsilon\log\epsilon)\right]\,, \quad \text{where:} \quad \mathcal{X} = \frac{1}{2}\int\frac{(\boldsymbol{x}\times\boldsymbol{\omega})\cdot\mathcal{I}}{|\mathcal{I}|^2}\,\boldsymbol{x}\,dV \tag{6}$$

is the vorticity centroid of the ring and $\beta_0$ is a numerical constant. This expression is valid in the small time limit as $\epsilon \to 0$ and assuming that the cross section remains Gaussian to first order. In the numerical work of Stanaway et al. (1988) the error bounds $\mathcal{O}(\epsilon\log\epsilon)$ derived by Saffman were found to be conservative, and the stronger bound of $\mathcal{O}(\epsilon^2\log\epsilon)$ was observed. Subsequent analytical investigations by Fukumoto and Moffatt (2000) confirmed these numerical results and provided improved assessments for the early time evolution of the ring velocity in the low and high Re limits (Fukumoto, 2009). For the low Re limit, the expression is as follows:

$$\mathcal{U}_F = \frac{d\mathcal{X}}{dt} = \frac{\Gamma_0}{4\pi R}\left[\log\frac{4R}{\sqrt{\nu t_\Gamma}} - \beta_0 - 4.5\left(\log\frac{4R}{\sqrt{\nu t_\Gamma}} - \beta_1\right)\frac{\nu t_\Gamma}{R^2}\right] \tag{7}$$

where $\beta_1$ is a numerical constant and $t_\delta = t + \delta^2/4\nu$ is a modified time factor.





### 3.1.2 Numerical Parameters

The vorticity field is initialized with the distribution given in Eq. (5) and on the domain with extent $X \in [-b\delta, X_u]$ and $Y, Z \in [-R_\delta, R_\delta]$ where $b$ is a buffer factor and $R_\delta = R + b\delta$ is chosen to ensure the vorticity distribution has compact support within $\mathcal{D}$. $X_u$ is chosen based on the length of the simulation to ensure the translating ring and it's wake remain within $\mathcal{D}$ for the
duration of the simulation. The uniform mesh spacing $h = h_x = h_y = h_z$ is varied to investigate the influence of grid size on accuracy. The particle set is advanced at each time step with a 4$^\text{th}$ order explicit Runge-Kutta scheme. The simulation is remeshed after every fifth time step and the time step $dT$ specified such that the cumulative Courant-Friedrichs-Lewy (CFL) number is below 1. The $M_2$ routine has been applied for the mapping of particle properties to and from the Lagrangian grid and during particle set remeshing. A fourth-order central FD scheme has been applied to calculate grid gradients. The free-
space Green's function chosen for the solution of the Poisson equation on the grid has a 4$^\text{th}$ order Gaussian regularisation for consistency with the FD scheme chosen. Magnitude filtering has been applied with a factor of $f_{mag} =$ 1e-05. Both magnitude filtering and particle set reprojection are carried out after remeshing.

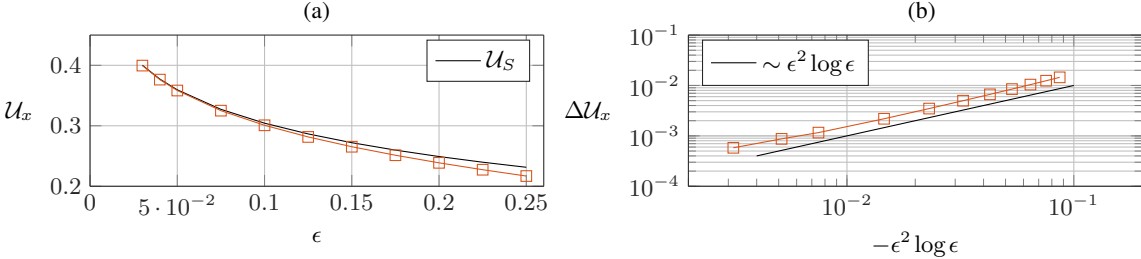

**Figure 5.** Translation velocity of the vorticity centroid of a vortex ring. (a) Velocity as a function of core size parameter. (b) Velocity error $\Delta \mathcal{U}_x = (\mathcal{U}_x - \mathcal{U}_S) R_0 / \Gamma_0$.

### 3.1.3 Steady Case- Initial Ring Velocity

The initial translational velocity field of the ring as a function of core size $\delta$ will be investigated. The ring has been initialised
with Re$= 100$. For each value of $\delta$, the grid size has been set to $H = \delta/12$ in order to ensure that the core is resolved. $X_u = 2b\delta$ has been set large enough that the entire ring cross-section is contained, and a small buffer provided for initial translation. The results are compared to the expression derived by Saffman in Fig. 5 for a range of initial core sizes. It can be seen that the translation velocity behaves approximately as the analytical solution due to Saffman- Eq. (6), with an error which grows with increasing $\epsilon$. Furthermore, the error in the predicted velocity aligns well with the improved error bound $\epsilon^2 \log \epsilon$ derived by
Fukumoto and Moffatt (2000).





### 3.1.4 Unsteady Case- Ring Evolution

The accurate time evolution of the vorticity field can be monitored by inspecting linear flow invariants. For a viscous flow field in the absence of a non-conservative forces, the total circulation $\mathcal{C}$ and hydrodynamic impulse $\mathcal{I}$ are conserved:

$$\mathcal{C} = \int \boldsymbol{\omega} \, dV \,, \quad \mathcal{I} = \int \boldsymbol{x} \times \boldsymbol{\omega} \, dV \,, \tag{8}$$

where $\boldsymbol{x}$ is the position of the particle and the volume integral is taken over all space. These quantities can be monitored during a simulation by numerically integrating the quantities discretely over the Eulerian grid $\mathcal{G}_E$- these integrals shall be referred to as flow *diagnostics*. Although the integrals above are performed over $\mathbb{R}^3$, the domain is chosen to have compact support of the vorticity field such the integrands in Eq. (8) vanish near the boundary and outside of the domain, making the evaluation of the integrals possible. This is done for the case of the translating ring for a range of grid resolutions and the results are displayed in

Fig. 6. Due to the symmetry of the vortex ring, the integral of the total circulation described by Eq. 5 should vanish. It can be seen that the total circulation remains very near zero for the duration of the simulation for the simulated grid resolutions. The magnitude of the error for the $H = \delta/3$ case suggests that the deviation is likely simply due to numerical round-off errors. It should be noted that for these simulations the solver was compiled to have double floating-point precision in order to illustrate the accuracy as these values are much larger for single floating-point precision. Inspection of the hydrodynamic impulse $\mathcal{I}$

demonstrates that as mesh resolution increases, this value converges towards that expected for a conserved field. These results demonstrate that in terms of linear diagnostics, the solver is functioning satisfactorily. In addition to the linear diagnostics

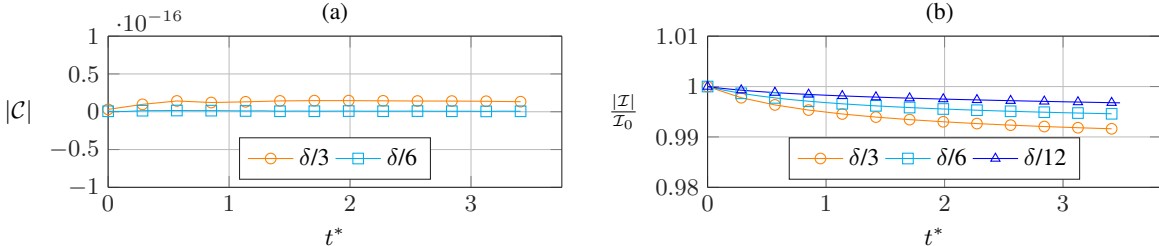

**Figure 6.** Evolution of the integral circulation $\mathcal{C}$ (a) and hydrodynamic impulse $\mathcal{I}$ (b) for the case of the translating vortex ring for three grid resolutions.

described above, two quadratic diagnostics can be specified- the total kinetic energy $\mathcal{K}$ and the enstrophy $\mathcal{E}$:

$$\mathcal{K} = \int \boldsymbol{u} \cdot (\boldsymbol{x} \times \boldsymbol{\omega}) \, dV \,, \quad \mathcal{E} = \int \boldsymbol{\omega} \cdot \boldsymbol{\omega} \, dV \,. \tag{9}$$

The time rate of change of $\mathcal{K}$ is related to the enstrophy through the relationship $\frac{d\mathcal{K}}{dt} = -\nu\mathcal{E}$ (Lamb, 1945). This can be used

as a means to track the accuracy of the time evolution of the system as described by Stanaway et al. (1988). These quantities have also been integrated during evolution of the vortex ring. The rate of change of the kinetic energy was calculated using central finite differences on the diagnostic $\mathcal{K}$. The evolution of the quadratic diagnostics is displayed in Fig. 7. The evolution of the quantity $-\nu\mathcal{E}$ is also shown for the finest resolution $H = \delta/12$ for comparison. It can be seen that as the mesh resolution





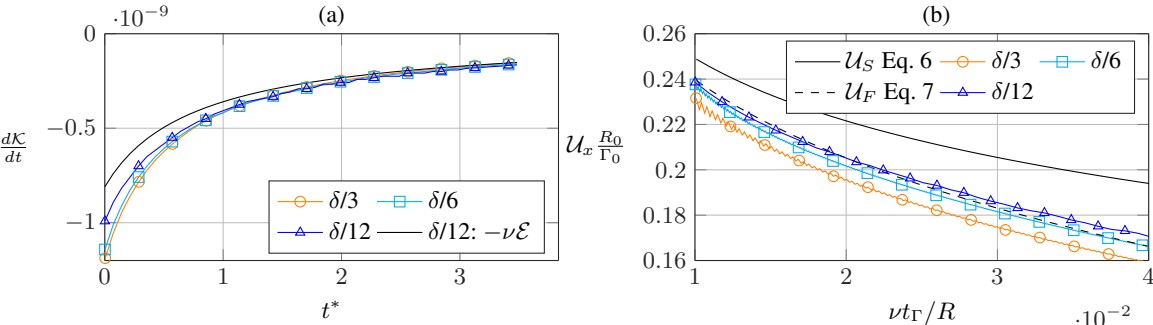

**Figure 7.** (a) Evolution of the rate of change of the integral kinetic energy $\mathcal{K}$ for the case of the translating vortex ring. (b) Translation velocity of the vorticity centroid as a function of normalized time.

increases, the rate of change of kinetic energy approaches the expected value. The evolution of the normalised translational

velocity of the vorticity centroid- $\mathcal{U}_x R/\Gamma_0$ is also displayed in Fig. 7. The value as predicted by the original theory by Saffman $\mathcal{U}_S$ (6) for arbitrary Re is shown along with the asymptotic version derived by Fukumoto for low-Re cases $\mathcal{U}_F$ (7) for reference. It can be seen that in the low resolution case a sawtooth pattern is observed in the evolution of $\mathcal{U}_x$. This is the result of both i) the particle redistribution process and ii) the mapping scheme, for both of which an explanation is readily available: When the grid resolution is increased, the time resolution must also be increased in order to satisfy the CFL constraint. As the global

time constant $t_\Gamma$ of the flow remains unchanged, smaller time steps between particle redistribution implies that the particle displacement on $\mathcal{G}_L$ is decreased, explaining point i). Equivalently, an increased grid resolution also implies that error in flow field moment conservation which is incurred during redistribution (here the $M_2$ scheme) are reduced, explaining point ii). It can be seen that as the mesh resolution is increased, the time evolution of the $\mathcal{U}_x$ agrees better with the expression due to Fukumoto $\mathcal{U}_F$. These results appear to be well aligned with similar investigation in the literature (Liska and Colonius, 2015).

As the expressions for $\mathcal{U}_S$ and $\mathcal{U}_F$ are valid in the limit $\epsilon \to 0$, this error is expected to increase as $t_\Gamma$ increases, as indeed is observed. These simulations demonstrate that for the case of an unbounded vortical flow, the solver is functioning correctly.

## 3.2    Lifting-Line: Symmetric Wing

In order to validate the solver for the case of an immersed lifting body, a wing with an elliptical planform has been simulated as analytical solutions are available for comparison. In adddition, the wing exhibits a continuous, well defined circulation

distribution. Following the description in Sect. 2.5, the NLL method has been applied to model the circulation distribution.

### 3.2.1    Known Solutions

The lifting line theory has been successfully applied to a range of lifting surfaces problems. Some of these results will be described here. The effect of lift generation on the flow field is modelled as a vortex filament which passes through the aerodynamic centre of the wing sections. The wake is composed of a superposition of horseshoe vortices which extend to





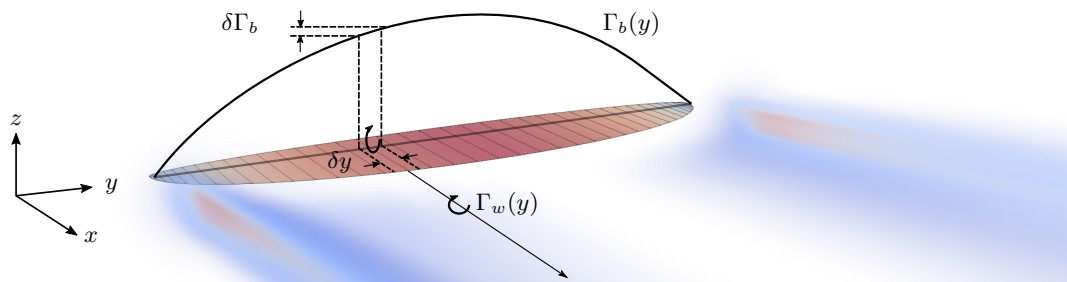

**Figure 8.** A visualisation of the lifting line method for an elliptical wing section. The spanwise gradient of the bound circulation $\Gamma_b(y)$ determines the strength of the wake vorticity $\Gamma_w(y)$.

infinity and have strength equal to the spanwise gradient of the circulation $\frac{d\Gamma}{dt}(y)$. This is visualised in Fig. 8. By applying thin airfoil theory for the calculation of the lift coefficient and integrating the effect of the downwash produced by the wake $\epsilon(y)$, Prandtl's integral equation for the circulation distribution along the lifting line can be derived:

$$\Gamma(y) = \frac{1}{2}a_0 c(y)[U_\infty \alpha_0(y) + \epsilon(y)] \text{ , where: } \epsilon(y) = \frac{1}{4\pi}\int_{-S/2}^{S/2}\frac{1}{y-y'}\frac{d}{dy'}\Gamma(y)\,dy' \text{ .} \tag{10}$$

In this equation, $a_0$ is the slope of the airfoil lift coefficient, $\alpha$ is the effective sectional angle of attack, $c(y)$ is the chord
length of the airfoil, $S$ is the span of the wing and $U_\infty$ is the freestream of the flow. As the circulation at the boundary of the body must vanish, a suitable approach for a symmetric circulation distribution is to assume a trigonometric series solution $\Gamma_{ts}$ of the form:

$$\Gamma_{ts}(\theta) = 2bU_\infty \sum_{1}^{N} A_j \sin j\theta \text{ ,} \tag{11}$$

where $y = \frac{S}{2}\cos\theta$. Substitution of this expressions into Eq. (10) leads to a linear system which can be solved for the coefficients
$A_j$ (Anderson, 2016). This is a convenient and simple method for numerically calculating solutions to Eq. (10) for arbitrary planforms and discretisations.

### 3.2.2 Nonlinear Lifting Line Method

Following the description given in Sect. 2.5.1, the model applied here is substantially the same as that described by Van Garrel (2003). Prior to inspecting the cases where the VPM method has been applied to model the wake, it is instructive to first validate
the NLL model. Two comparative cases have been considered: a wing with an elliptical planform and a wing with a rectangular planform. For both cases investigated, a steady system is modelled with the wake extending from the airfoil trailing edge to a distance $100\,S$ downstream of the wing. For all cases, the wing has been discretized with $N$ elements with cosine spacing and cosine shifting of the control point, a so-called full-cosine spacing as described in Van Garrel (2003). The first case inspected





is that of an elliptical planform such that $c(y) = c_0 \sqrt{1 - 4y^2/S^2}$ with root chord length $c_0 = 1$ m and span $S = 5$ m. In this
case an analytical solution for the downwash can be found from Eq. (11):

$$\epsilon_{ell}(y) = -\frac{\Gamma_0}{S} \text{ , yielding an elliptical circulation distribution: } \Gamma_{ell}(y) = \Gamma_0 \sqrt{1 - \left(\frac{2y}{S}\right)^2} \qquad (12)$$

The NLL model has been applied with under-relaxation factor of 0.2 and an initial angle of attack $\alpha_0 = 5.7106^o$. The case
$N = 40$ is shown initially to demonstrate the accuracy of the model. The circulation distribution using the NLL model are
shown in Fig. 9. For comparative purposes, the analytical result of Eq. (12) is shown along with the trigonometric series solution
$\Gamma_{ts}$. In the case of the latter and in order to maintain consistency with the NLL model, the blade has been discretised with 40
blade elements with a cosine distribution. It can be observed that for the majority of the blade span, the circulation predicted
by the NLL method is indistinguishable from the analytical solution, as is the result of the trigonometric series solution $\Gamma_{ts}$.
In order to provide more detail, the region $0.45 \leq y/S \leq 0.50$ has been magnified as here the gradient $\frac{d\Gamma}{dy}$ is strongest. In
this region, almost no disparity between the models is observable. The analytical value for the induced downwash of the
elliptical wing- Eq. (12) is also shown in Fig. 9. For the case of $\alpha = 5.7106^o$ this value is approximately $\epsilon/U_\infty = -0.0238$.
The downwash predicted by the trigonometric series solution $\Gamma_{ts}$ can easily be evaluated as a product of the $A_n$ as carried out
in Anderson (2016). Additionally, by numerically integrating the effects of the wake vortex filaments, the induced downwash
$\epsilon_{nll}$ can also be calculated for the NLL model and compared to the analytical and numerical values as was done for the
circulation distribution. As with the circulation distribution, over the majority of the span the differences in induced velocity
are indiscernible from the analytical and trigonometric series solutions. In the tip region, the effect of the discretisation of the
NLL model can be observed and $\epsilon_{nll}$ slightly deviates from the analytical solution.

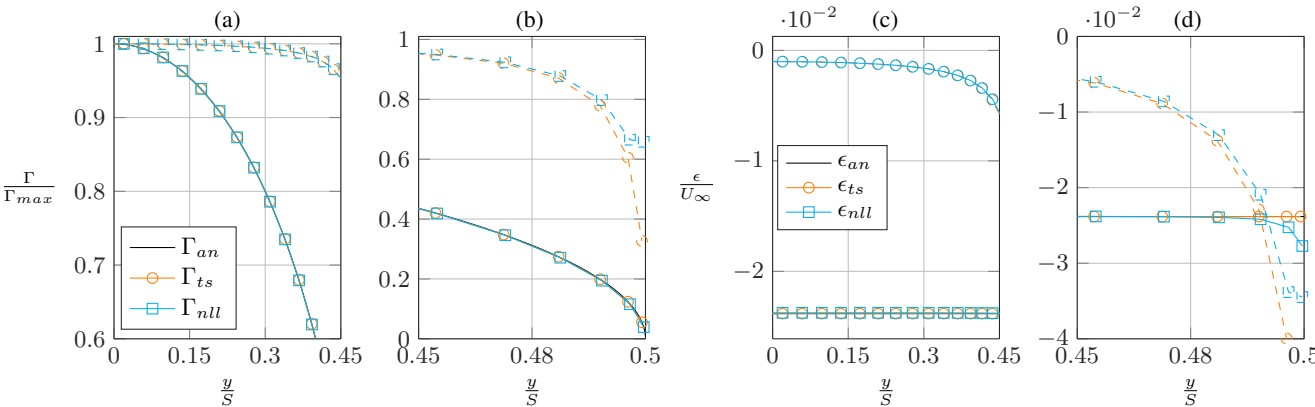

**Figure 9.** Numerical solutions for an elliptical (solid) and rectangular (dashed) wing. Predicted circulation distributions are shown in plots
(a) and (b). Predicted downwash values are shown in plots (c) and (d). The analytical results $\Gamma_{an}$ are calculated with Eq. (12) for the elliptical
wing. The numerical results $\Gamma_{ts}$ is calculated with the trigonometric series method of Eq. (11). $\Gamma_{nll}$ is calculated by using the NLL method.

For the case of a rectangular wing such that $c(y) = c_0$, an analytical solution for the circulation distribution has been derived
by Stewartson (1960), however this solution is only valid in the limit of infinite span $S$ and is therefore of limited application to



practical cases. In the proceeding, comparison is therefore made only to the solution obtained using trigonometric series $\Gamma_{ts}$. It

has been observed that the solution of a rectangular wing, owing to the constant chord length has a much stronger interaction with the tip vorticity distribution. This poses numerical challenges for the NLL model, which can only partially be alleviated through the use of the vortex core model. For this reason, a wing with chord $c_0 = 1.0$ m and span $S = 100$ m has been chosen to ensure that the wing has a very high aspect ratio. In addition, the $\delta$-style core model of Van Garrel (2003) has been used, with $\delta = 0.05$. As with the elliptical wing case, the rectangular wing has been discretised with $N = 40$ elements. The results

comparing the two methods of calculation for the circulation distribution are shown in Fig. 9. It is again observed that over the majority of the wingspan, the deviation between $\Gamma_{ts}$ and $\Gamma_{nll}$ is negligible with the exception of the tip region, where the disagreement is stronger than for the elliptical planform case. Inspection of the downwash induced by the wake, a similar trend is also observed. These tests demonstrate that the NLL model indeed produces accurate results and can be applied to determine the circulation of lifting bodies and their wake signature for the VPM method.

### 3.2.3    Steady Case- Ideal Wake Distribution

The validation of the VPM model will now be carried out by comparing the velocity field induced by the VPM model to the vortex filament wake system of the standard NLL method.

The velocity field induced by wake filaments along with the wake circulation distribution are updated during the fixed-point iteration. These values are then extracted after convergence of the NLL method. In the case of the VPM model, wake

filaments calculated by the aforementioned procedure are converted to a corresponding particle set and are mapped to the Eulerian grid $\mathcal{G}_E$ with the $M_2$ mapping procedure. The velocity field is then calculated through the procedure defined in Sect. 2.2 using a 4th-order Gaussian kernel. The velocity is then sampled at the corresponding on-blade coordinates using the $M_2$ mapping procedure. This has been done for three grid discretisations: $H = S/N$ where $N$ equals 20, 40 and 80. In each case, the VPM domain was generated to be large large enough to contain all wake elements. This corresponds to $\mathcal{D} =$

$[-B, 100S + B] \times [-(B + 0.5S), B] \times [-B, B]$, where $B = 10H$ is a boundary buffer factor. The results for these comparisons are shown for the elliptical and rectangular wings in Fig. 10. The tip region has again been magnified to emphasise differences between the models.

Inspecting the result for the elliptical wing, it can be seen that the downwash predicted by the filament method is very near the analytical result. In comparison, the VPM method still deviates for all resolutions in the tip region. However, it can be

directly observed that as N increases, the error over the entire span decreases. The increased error compared to the filament method can be understood as the result of two factors. The first factor is the spatial discretisation. The filament method resolves both the blade and the wake filaments with a cosine spacing, implying that the induced velocity field is better resolved in the tip region. In comparison, the VPM method resolves the velocity field on a grid with homogeneous resolution. The second factor is the form of regularisation applied. In the filament method, as described in the previous section, no regularisation is active for

the elliptical wing. This implies that the solution corresponds directly to a distribution of potential flow vortex filaments in the wake. This is a key underlying assumption of the solution procedure for $\Gamma_{ts}$. In comparison, the VPM method uses a Gaussian regularisation for the calculation of the velocity field in order to avoid singularities in the solution field. This strongly modifies





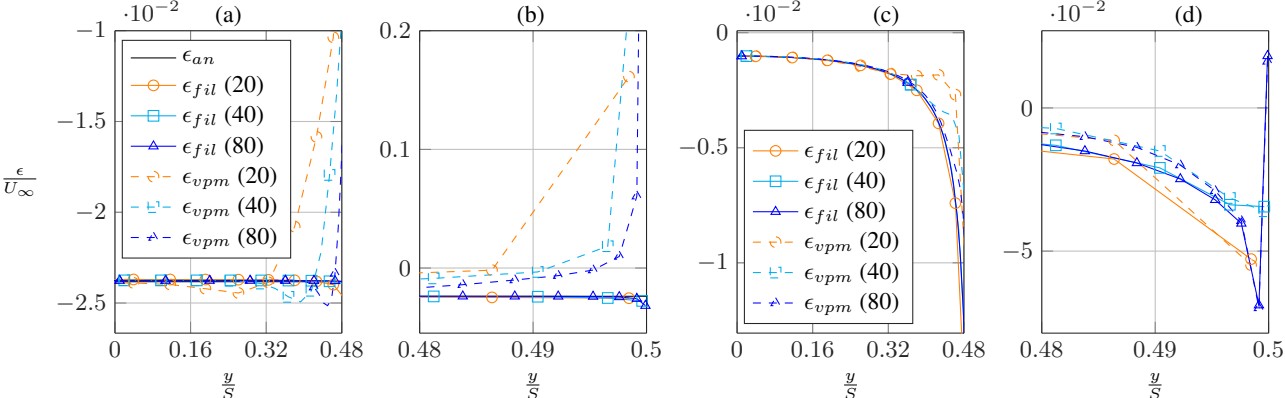

**Figure 10.** On-blade downwash as predicted by the filament wake model and the VPM model within. Plots (a) and (b) show results for the elliptical wing. Plots (c) and (d) show results for the rectangular wing. The analytical results $\epsilon_{an}$ refers to Eq. (12) for the elliptical wing. $\epsilon_{fil}$ results obtained by using the filament method with (N) elements. $\epsilon_{vpm}$ are results obtained by using the VPM method with mesh resolution $H = S/N$.

the solution in the near-field of a point source and has the effect of mollifying the velocity field. For a detailed investigation of this topic the reader is referred to Caprace et al. (2018), where similar effects where observed and and a range of 1D, 2D, and 3D regularisations were investigated. The error introduced by both of these factors is expected to decrease as resolution increases, which is indeed observed in Fig. 10.

In the case of the rectangular wing, no analytical solution exists. Comparison is therefore made directly to the NLL solution $\epsilon_{fil}$. Similar results can be observed as with the elliptical wing, however the deviations are stronger between the two models. In addition to the discretisation error described previously, the regularisation is modified. The $\delta$-method described in Van Garrel (2003) has been used. As opposed to the Gaussian kernel, the $\delta$ regularisation simply removes the singularity in a comparable fashion to a Rankine vortex model. Despite the difference in regularisation models, the results here again appear to be similar to those for the elliptical wing in that increasing resolution of the VPM model converges towards the solution for the $\delta$-model applied for the filaments. These results demonstrate that for a sufficiently finely resolved grid discretisation, the VPM method converges towards accurate results for the induced velocity field when filament-particle conversion is carried out.

### 3.2.4 Unsteady Case- Wake Rollup

In order to validate the VPM model for the case that the vorticity field is evolving in time, the wake of an elliptical wing is simulated. Kaden (1931) investigated the asymptotic behaviour of the vortex sheet shed by a wing with an elliptical circulation distribution. The flow field is assumed to be inviscid and to behave approximately as a 2D field in the far wake. Under these assumptions, the vorticity at the wing tips rolls together and merges into two single, counter-rotating vortex pairs in the far wake. Kaden demonstrated that the lateral displacement (spanwise) between the vortex pairs converges to the value $C_y = S\pi/8$. This





**Table 1.** Simulation parameters for unsteady VPM simulations of an elliptical wing.

| H [m] | dT [s] | $C_y$ [m] | | $\epsilon_\infty$ [ms$^{-1}$] | |
|---|---|---|---|---|---|
| | | Value | Rel. Error (%) | Value | Rel. Error (%) |
| $S/40$ | 0.015 | 1.9174 | 2.35 | -0.0110 | 42.98 |
| $S/60$ | 0.01 | 2.0079 | 2.26 | -0.0166 | 13.95 |
| $S/80$ | 0.0075 | 1.9787 | 0.77 | -0.0191 | 0.99 |

vortex pair continues to translate under their own self-influence at a constant velocity. Kaden was able show that this velocity converges to the value of $\epsilon_\infty = \epsilon_{ell}8/\pi^2$ where $\epsilon_{ell}$ is the analytical downwash given by Eq. (12).

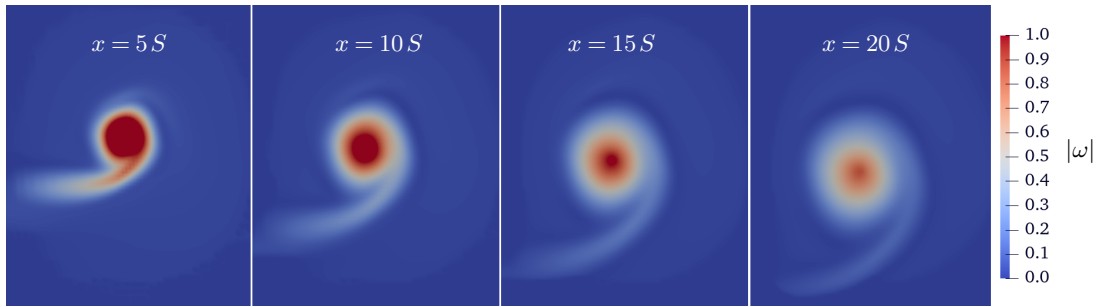

**Figure 11.** Planes of constant $x$ demonstrate the evolution of wing tip vortex in the wake of an elliptical wing. Only the region $y < 0$ is shown as the flow field is symmetric about the plane $y = 0$.

430    This scenario is modelled using the VPM solver by specifying the kinematic viscosity $\nu = 0$ in the solver. In order to reduce simulation length, the transients of the initial on-blade circulation are circumvented by taking the steady-state NLL solution $\Gamma_{nll}$ for the elliptical wing from the previous section directly to define the wake vorticity distribution. Mapping between $\mathcal{G}_E$ and $\mathcal{G}_L$ was carried out with the M$_2$ scheme. The system was integrated with a fourth-order Runge-Kutta timestepping scheme with a timestep dT, chosen such that the cumulative CFL number does not exceed unity. These values are summarized in

435    Table 1. At each timestep, a wake element corresponding to a filament with the circulation specified by $\Gamma_{nll}(y)\Delta y$ with length specified by the flow field advection $dL = dT \cdot U_\infty$ was generated for each spanwise station. This was then mapped to the Lagrangian grid $\mathcal{G}_L$ using the M$_2$ scheme. The VPM domain was generated to allow the wake to converge sufficiently such that the spanwise position of the vorticity centroid is constant. It was found during testing that a domain length of $40S$ was sufficient for this purpose. This corresponds to $\mathcal{D} = [-B, 40S+B] \times [-(B+0.5S), (B+0.5S)] \times [-B, B]$, where $B = 40H$

440    is again a boundary buffer factor.

The evolution of the wing tip vortex vorticity distribution for a range of downstream positions are shown in Fig. 11. It can be observed that alone through the process of vortex stretching, the vorticity distribution spreads and forms a single strong vortex. The non-axisymmetric nature of this vortex core is also visible. It is also seen that the vortex pair continue to translate



downwards underneath their own self-induction in the far wake. The $y$-position of the vorticity centroid is also seen to shift
inboard, as per the analytical prediction of Kader. In order to quantify the accuracy of the solver, the value of the $C_y$ was
extracted from the converged wake by taking a plane $x =$const.$= 20D$ downstream of the wing and calculated the position of
the vorticity centroid. Additionally, the average velocity field in $z$ direction is sampled to calculate the wake downwash $\epsilon_\infty$.
These values have been summarised in Table 1 along with their relative errors compared to the analytical solutions of Kader.
It can again be observed that as the grid resolution increases, the values approach those predicted analytically by Kaden. The
nature of the convergence however appears to demonstrate that the velocity field is much more sensitive to the discretisation
then the evolution of the particle field. The metric $C_y$ is deceptive as the relative motion of the tip vortex is small. Additionally,
a number of factors influence the calculation of $\epsilon_\infty$, making exact identification of error sources challenging. The course grid
size influences the solution of the $\Gamma_{nll}$, as described in the previous section, along with the expected error of the integration of
the velocity field due to the use of a regularised kernel. Furthermore, the errors caused by the use of FD stencils to calculate
gradients in the flow field has a compounding effect on the error as the solution is time-marched. Further tests may be carried
out to demonstrate the convergence behaviour of the velocity and vorticity gradient fields. For brevity, this is left for further
work on winged devices. These simulations demonstrate that for a lifting body, the VPM model can be effectively used to
calculate the evolution of the velocity field along with the evolution of the wake of a lifting body.

### 3.3 Lifting-Line: Rotor

The VPM model will now be applied to the simulation of a rotor. Initially the model will be validated against an analytical rotor
model, the Betz rotor. Following this, a comparison to the Mexnext experimental campaign will be carried out to validate the
model against measurements made of an experimental rotor (Schepers et al., 2014). For the following analyses it is instructive
to introduce the tip speed ratio of the rotor, which relates the tip velocity of the rotor to the inflow velocity. This is given by
$\lambda = R\omega/U_\infty$ is a where R is the tip radius of the rotor, $\omega$ is the rotational velocity and $U_\infty$ is the undisturbed inflow velocity.

### 3.3.1 Known Solutions

A number of rotor solutions exist including Joukowski, Betz & Glauert rotors as described in Sørensen (2016). These models
assume helical symmetry of the wake structure of the rotor. For a helical wake topography, the helix can be parametrised with
$t$ as:

$$\boldsymbol{x}(t) = r\cos t\,\boldsymbol{e}_x + r\sin t\,\boldsymbol{e}_y + lt\,\boldsymbol{e}_z \,, \tag{13}$$

where $r$ is the radius of the helix and $h = 2\pi l$ is the pitch of the helix. Using the assumption of a rigid helix, the pitch can be
related to the tip speed ratio by $h = 2\pi R/\lambda$.

### 3.3.2 Steady Case: Betz Rotor

Owing to the availability of analytical solutions, the Betz rotor has been chosen to demonstrate the steady rotor solution with
the VPM solver. The theory of the Betz rotor will briefly be described, followed by a validation of the velocity field using the



VPM method. Using kinematical considerations of a rigid screw, the boundary condition for the axial velocity component on the helical surface of a Betz rotor can be specified as (Sørensen, 2016):

$$u_x(r) = U_\infty \frac{r^2}{r^2 + (Rh)^2} \ .$$

(14)

An analytical solution for this was derived by Goldstein and Prandtl (1929) which uses a series solution employing Bessel functions. A range of approximations exist for this problem as described in Branlard (2017). A method suggested by Okulov and Sørensen (2008) makes use of the boundary condition (14) along with an expression for the velocity induced by an infinite helical vortex filament. This expression has been omitted for brevity, however the reader may find this in Branlard (2017). This approach greatly reduces the complexity of the problem as the task is reduced to the setup and solution of a relatively simple linear system. Solutions for a range of single- and multi-bladed rotors are shown in Fig. 12. For the simulations carried

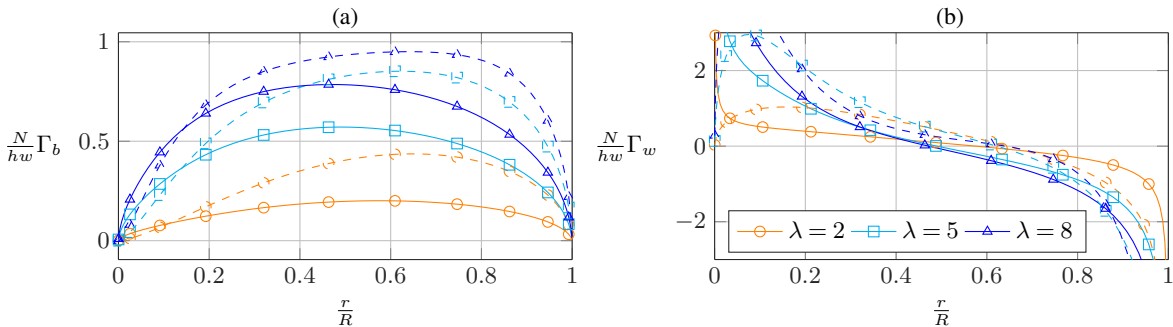

**Figure 12.** Circulation distributions for the Betz rotor for 1-bladed (solid) and 3-bladed configurations (dashed). (a) On-blade circulation distribution. (b) Wake circulation distribution.

out here, the rotor blade is discretised into $N$ sections with radial position $r_i$ and width $\Delta r_i$. The aforementioned approach has been used to calculate the wake circulation distribution as a function of position $\Gamma_b(r_i)$. For each radial section, the arc of a helix is generate which uses the parametrisation of Eq. (13). The arclength of a single winding of this helix is given by $2\pi\sqrt{r_i^2 + h^2}$. $D$ windings of the helix are taken such that a sufficient section of the (semi-infinite) wake is treated. This filament is then discretised into segments of length $H$, the characteristic grid size of the VPM solver. Each segment is then converted into a particle as was done in the previous sections. This process is repeated for all radial sections. Through this procedure, the analytical wake circulation distribution is discretised and can be mapped to the Eulerian grid of the VPM solver. The velocity field is then calculated using the procedure of Sect. 2.2 and the result is compared to the analytical solution for the inflow velocity given by Eq. (14). This has been carried out for a 3-bladed rotor operating at $\lambda$ tip speed ratios of 2 & 8. The results are displayed in Fig. 13.

The results shown in plot (a) demonstrate the effect of increasing the grid resolution. In this case the 4[th]-order Gaussian regularisation has been used. It can be observed for both $\lambda$ values that increasing the grid resolution reduces the error as the solution approaches the analytical distribution given by Eq. (14). The discrepencies observed in the tip region, following a similar argument made in Sect. 3.2, is a result of two factors. The first factor is the homogeneous grid resolution being of lower





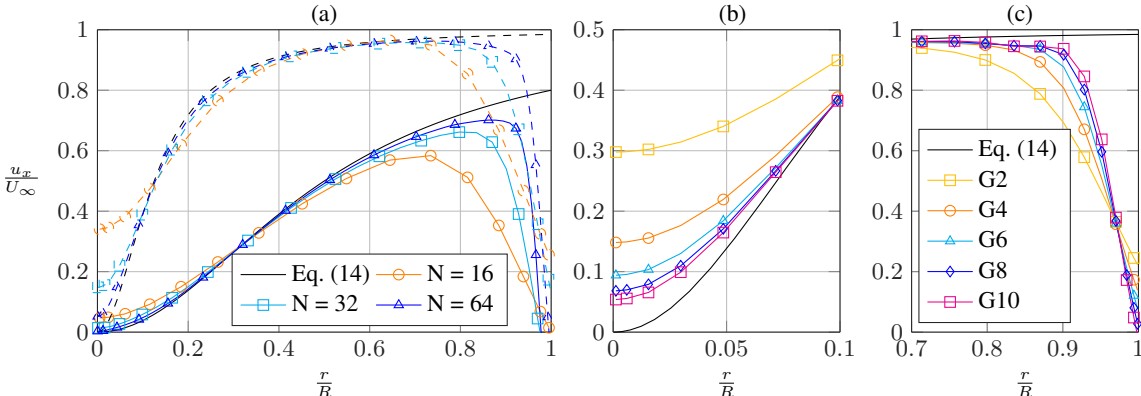

**Figure 13.** Axial velocity induced by the wake of a Betz rotor at different radial positions of blade 1 for a range of grid discretisations. (a) Axial velocity for the case $\lambda = 2$ (solid) and 8 (dashed). (b) & (c): Axial velocity for the case $\lambda = 8$ with a range of Gaussian regularisation functions.

resolution than the tip resolution of the numerical Betz model, which has a cosine discretisation and can therefore better resolve the gradient at the tip. The second factor is the effect of the regularisation function. This factor is investigated in plots (b) and 500  (c) where the grid discretisation is held constant with $N = 32$ and the impact of the regularisation function on the solution is investigated. The root and tip sections are magnified in plots (b) and (c), respectively. A range of Gaussian regularisations have been investigated from 2nd order (G2) up to 10th order (G10). It can be seen that as the order of regularisation is increased, the result approaches the analytical solution given by Eq. (14). It can also be seen that the regularisation G10 produces a discontinuity at the position $r/R \approx 0.83$. This has been observed in other cases where grid resolution is low and is a result of 505  the insufficient smoothness of the input field. This phenomena was not seen to occur for the G10 resolution in the validation of the `SailFFish` solver as an input field with $C^\infty$ smoothness was used. The results indicate that the VPM solver accurately resolves the case of a steady helical rotor wake.

### 3.3.3 Unsteady Case: Mexnext Rotor

The validation of the VPM/NLL models for a representative horizontal axis wind turbine architecture has been carried out by 510  simulating the experiments carried out in the Mexnext experimental campaign (Schepers et al., 2014). The Mexnext campaign was a follow-up to the **M**odel **Ex**periments **I**n **Co**ntrolled Conditions (MEXICO) campaign (Schepers and Snel, 2007). A comprehensive dataset of turbine and wake data was collected over a range of operating conditions. This dataset includes a variety of on-blade pressure and load measurements along with wake velocity measurements and represents therewith one of the most complete openly available datasets for experimental turbines. The experimental setup will first be described followed by a 515  description of the numerical model. Following this, the results of the VPM model at three levels of resolution are compared to a vortex filament type wake model along with the experimental results at three tip speed ratios.



The Mexnext rotor is a three-bladed horizontal axis wind turbine with a diameter of 4.5 m ($R_{tip}$=2.25 m). The experiments were carried out in the open-jet wind tunnel of the **L**arge scale **L**ow speed facility (LLF) of the German Dutch Wind Tunnels (DNW). The turbine was operated at a constant rotational speed of $424.5$ RPM at three inflow velocities $U_\infty$: 10.0, 15.1 and 24.1 ms$^{-1}$, corresponding to tip speed ratios $\lambda$ of 10.0, 6.7 and 4.2, respectively. Experimental results from Phase III of the Mexnext campaign have been used for the comparison herein. Only operational states with zero yaw are considered for this validation as yawed cases would require the implementation of dynamic stall model due to the rapidly varying angle of attack. This is considered as future work for the aerodynamics module. The MEXICO rotor blade is comprised of three sections, each with a corresponding airfoil profile: the DU-91-W2-250 foil (0.2-0.46 $R_{tip}$), the RISØ A2-21 airfoil (0.54-0.66 $R_{tip}$) and the NACA 64-418 airfoil (0.74-1.0 $R_{tip}$). Airfoil polars for these sections were provided for the Mexnext blind tests and these have been utilized as input to the NLL model for this analysis.

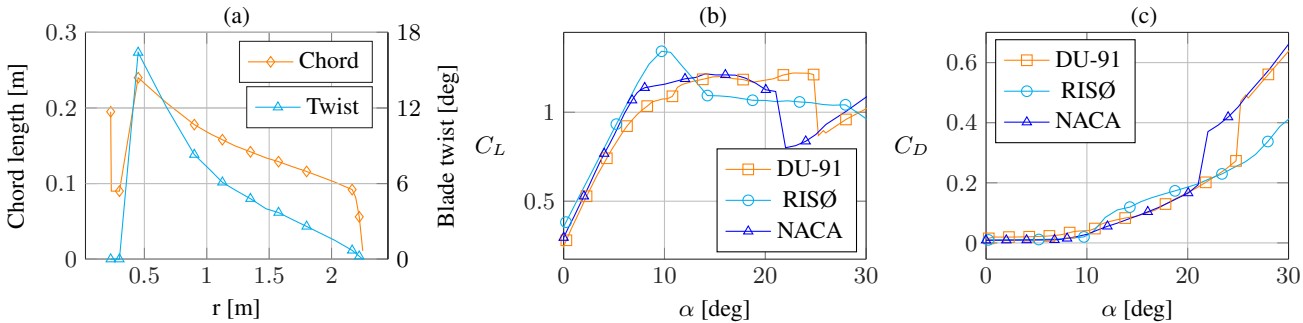

**Figure 14.** Provided MEXICO rotor blade geometry and aerodynamic data. (a) Blade twist and chord length. (b) Lift coefficients. (c) Drag coefficients.

Three operating conditions have been simulated using the VPM method with a NLL model. The blade has been discretised with N panels and a full-cosine discretisation. The blade has been truncated at r = 0.21 m as the root region is composed of non-lifting cylindrical sections which contribute negligibly to aerodynamic loads. The MEXICO blade chord and twist distribution along with profile aerodynamic lift and drag coefficients have been used to generate the numerical model of the blade with the NLL model. This data has been provided for an aerodynamic blind test (Schepers et al., 2014). Blade geometry and airfoil lift and drag coefficients for a range of angles of attack are shown in Fig. 14. Airfoil coefficients have been interpolated from the bounding airfoil data for transitional spanwise sections.

Two wake models have been applied in the following analysis. The first model (FIL) uses only vortex filaments to treat the wake. The wake is allowed to deform through the calculation of the velocity induced on the connecting wake nodes, the motion of which is integrated with an Adams-Bashforth 2$^{nd}$-order integration scheme. The induced velocity is carried out with a direct evaluation, such that the problem scales unfavourably with an increasing number of wake elements. This approach is comparable to that described for the AWSM wake model in the comparative study (Schepers et al., 2014). As the turbine is unyawed, the turbine loads eventually converge along with wake. This implies numerically that the trailing circulation of the wake eventually asymptotes to zero strength. These have therefore been deactivated in order to allow for quicker convergence



**Table 2.** Simulation parameters for the filament wake (FIL) and particle wake (VPM) models.

| Case | H | $\Delta\theta$ [deg] | $N_{rot}$ | $X_W/R_{tip}$ | Filament reg. | Particle reg. | $f_{mag}$ |
|------|-----|---------------|-----------|---------------|---------------|---------------|-----------|
| FIL | D/40 | 5.0, 5.0, 5.0 | | | van Garrel | - | - |
| VPM1 | D/20 | 8.0, 6.0, 4.0 | | | van Garrel | G8 | 1e-4 |
| VPM2 | D/40 | 4.0, 3.0, 2.0 | 30, 20, 10 | 13, 12, 11 | van Garrel | G8 | 1e-4 |
| VPM3 | D/80 | 2.0, 1.5, 1.0 | | | van Garrel | G8 | 1e-4 |

of the wake and to reduce the number of wake elements. The second wake model (VPM) is the particle scheme described in Sect. 2. This makes use of the NLL model with a potential flow region. The simulation domain has been specified such that x,y $\in$ (-3$R_{tip}$, 3$R_{tip}$) and to ensure the grid contains the necessary length to contain the entire advected wake: x$\in$ (-$R_{tip}$/2 , $X_W$).

The particle set is remeshed after every 5 timesteps, after which magnitude filtering and particle set reprojection is carried out. The $M_2$ routine has been applied for field and particle mapping. A combined Adams-Bashforth ($\omega_i$) / Leapfrog ($d_i$) 2nd-order timestepping scheme has been used to integrate the particle set. Simulation parameters have been summarised in Table 2.

The first results inspected are the sectional normal loads on the rotor blade for the three operational points. These are plotted

together with the experimental results in Fig. 15 (a)-(c). The variation in the net magnitude of the normal force $F_n$ is captured well by all models and appears to be best captured for the 10 ms$^{-1}$ case by the highest resolution VPM grid. The sectional angle of attack for each operating point is shown in Fig. 16 (a), where it can be seen that for the 10 ms$^{-1}$ case, blade sections experience moderately low angles of attack, implying that the flow remains attached along most of the blade span. The loads at the blade root region generally appear to be badly captured for the 24 ms$^{-1}$ case. Within the lifting line model, the aerodynamic

interaction of neighboring blade sections and the generation of wake circulation are based on potential flow, which becomes less accurate when the flow separates and strong interaction with the boundary layer and wake turbulence occurs. This explains the deviation in the normal loading in the root region for this case as the airfoil here is stalled. The loading profile predicted in Fig. 15 (b) is very similar to that predicted by similar NLL models in the works of Boorsma and Schepers (2016) and Ramos-García et al. (2017). An extensive study comparing experimental results with a range of NLL solvers was carried out in

the Mexnext campaign (Boorsma et al., 2018). Comparison of the predicted results display excellent agreement with the other solvers for all operating points. It is seen that the predicted loads deviate from the experiment at the blade mid-span for the 15 & 24 ms$^{-1}$ cases. Fig. 16(b) demonstrates that these deviations are caused by local maxima (15 ms$^{-1}$) and minima (24 ms$^{-1}$) in the sectional lift and drag coefficients of the blade. In Fig. 14 (b), it is seen that the provided airfoil polars for the DU-91 and RISØ sections stall at angles of attack of approximately 10°& 25°, respectively. These stall angles are displayed in Fig. 16

(a) and allow for a clear interpretation of the model predictions. In the 15 ms$^{-1}$ case, the high $C_L$ regime of the RISØ section compared to the DU-91 section explains the jump in $C_L$ and normal loads in that region. This discrepancy was observed for a range of other NLL and CFD solvers in alignment with the results found here (Boorsma et al., 2018). The deviation in the results appear to stem from the provided airfoil polars, which display strong discontinuities in these regions. To illustrate this,





the Reynolds number of the provided airfoil polars along with the predicted values shown in Fig. 16 (c). It is seen that the RISØ section operates in a range unmatched to the provided airfoil data (Re = 1.6e6).

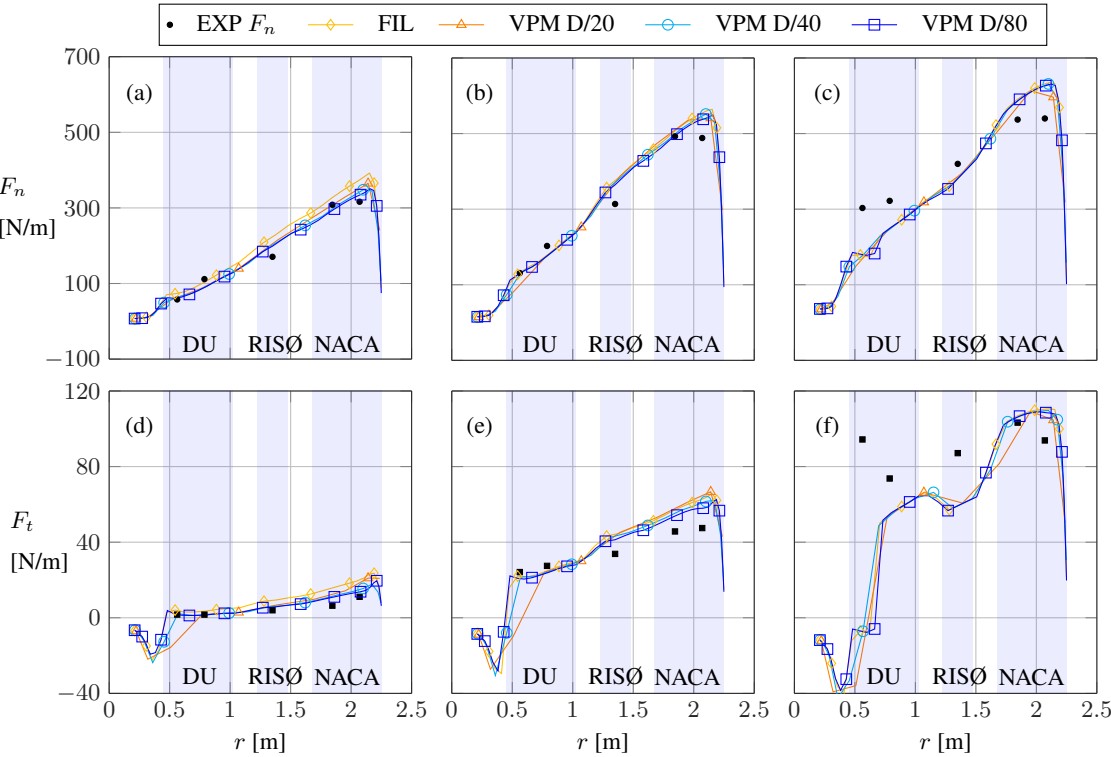

**Figure 15.** Rotor blade normal (top row) and tangential (bottom row) force coefficients from experiment, filament and VPM wake models for inflow speeds of $10 \text{ms}^{-1}$ (left column), $15 \text{ms}^{-1}$ (centre column) and $24 \text{ms}^{-1}$ (right column). Shaded areas represents airfoil sections.


It was suggested in Boorsma et al. (2018) that the disagreement in this region between the solvers and experiment may be due to either excessive boundary layer tripping in the RISØ section or the result of a strong camber change between the DU-91 and RISØ sections. For the purposes of the validation of the current solver, it suffices to conclude here that the results agree well with other solvers and that experimental deviations require detailed investigation of the experimental setup. As

seen in Fig. 16 (c) for the $24 \text{ ms}^{-1}$ case, the outer section of the DU-91 region remains attached, which transitions into a stalled RISØ region. This explains the strong drop in $C_L$ seen in Fig. 16(b) and the associated dip of the blade loads. The general agreement with experiment for the $24 \text{ ms}^{-1}$ case appears to be much worse than for the other operating points. In Fig. 16 (b) it is seen that the entire blade mid-span is stalled. As described above, the NLL is not expected to perform well for this case.

It can be seen in Fig. 16 (a) that the local angle of attack tends to increase in the tip region. This provides an explanation for the 15 & $24 \text{ ms}^{-1}$ cases, where all models tend to overpredict the tip normal force. This effect results from the induction field of the tip vortex. As this region is within the $\mathcal{D}_\Phi$ domain, the velocity induced by the tip vortex is treated directly by the vortex



filament method. It was seen in the analysis for the Betz rotor (Sect. 3.3.2) that the choice of regularisation strongly impacts the velocity field here. The impact of the choice of regularisation in the tip region was demonstrated in the work of Caprace
et al. (2018) and it should be expected that a more suitable choice of tip modelling would improve the result here. In addition, similar overpredictions of tip loading were found using other solvers, including finite volume solvers which resolve the blade geometry as described in Schepers et al. (2014).

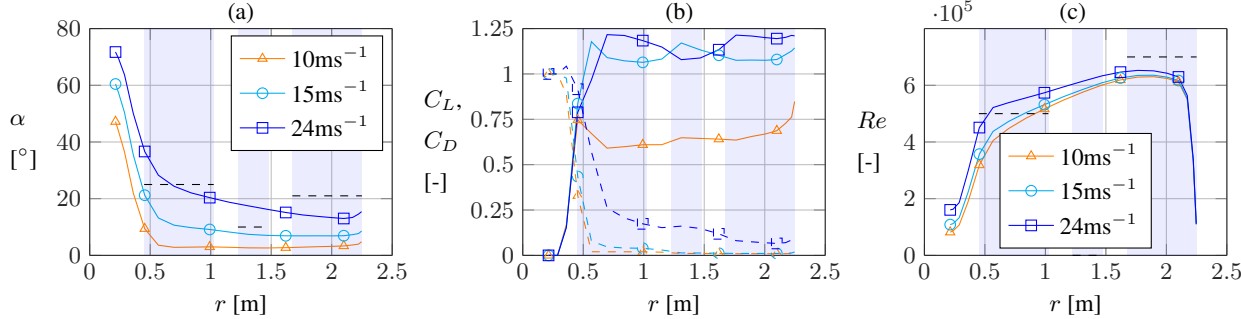

**Figure 16.** Blade sectional aerodynamic properties as predicted by the VPM solver with discretisation D/40: (a) Angle of attack: Sectional stall angle (dashed) is also shown. (b) Lift coefficient (solid) and drag coefficients (dashed). (c) Reynolds number: Provided polar $Re$ (dashed) also shown. Shaded areas represents sections of the DU-91, RISØ, and NACA profiles, respectively inboard to outboard.

The predicted tangential force coefficients are plotted together with the experimental results in Fig. 15 (d)-(f). The experimental agreement appear to be best captured for the case $10\,\mathrm{ms}^{-1}$, whereby the angle of attack along most of the blade section
is quite low and the underlying assumptions of the NLL model are met. In this region, the airfoil polars can be expected to be representative of the experimental behaviour as the blade loading is well-represented by the linear section of $C_L$ curve. The deviations at the blade mid-span for the $15\,\mathrm{ms}^{-1}$ follow directly from the previous discussion on the normal force coefficients. The strong disagreement for the $24\,\mathrm{ms}^{-1}$ case is again a result of the detached flow in the blade mid-span. The tangential force is seen to be somewhat captured by the NLL model in the tip region, as this region is not stalled as can be seen in Fig. 16
(a). The magnitude and distribution of the predicted tangential loads again agrees very well with the prediction of a number of other NLL solvers in Boorsma et al. (2018).

In summary, the quality of the results shown here are strongly dependent on the model input, in particular the provided airfoil polars. Furthermore, the experimental results demonstrate quite a high level of uncertainty, as described in Schepers et al.
(2014). Despite these points, the general trends and magnitude of the blade loading appears to agree well with the experiment and the predictions of the model are in good agreement with similar NLL and CFD solvers (Boorsma et al., 2018). In the next section, the velocity field in the wake of the rotor will be inspected. A number of hot-wire traverses were carried out in the Mexnext experiment to collect velocity data in the wake of the rotor. These allows for comparison between the experimental measurements and numerical predictions of the VPM and FIL models.






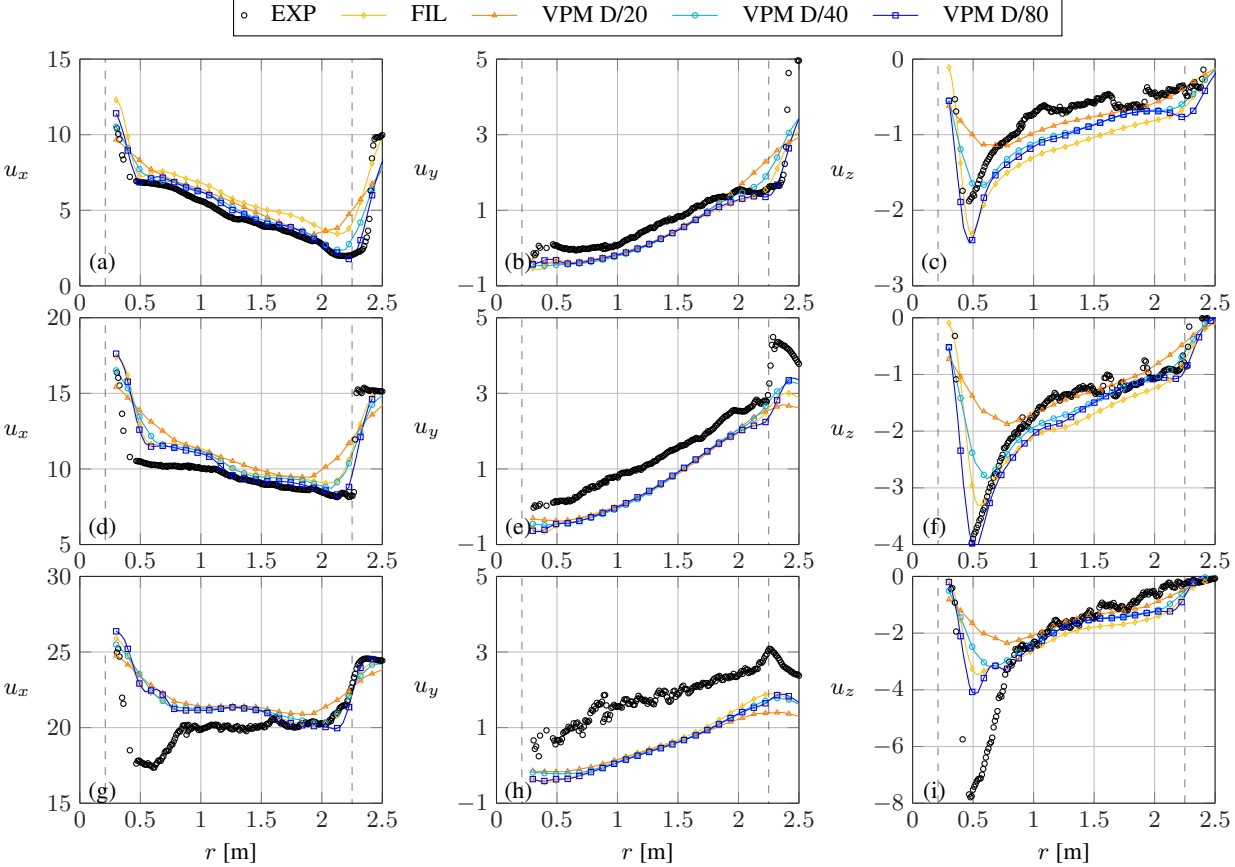

**Figure 17.** Experimental and numerically predicted velocities along a downstream radial traverse at the position x=0.3 m. Inflow velocity: 10 ms$^{-1}$ (top row), 15 ms$^{-1}$ (middle row), 24 ms$^{-1}$ (bottom row). Velocity components: Axial velocity $u_x$ (left column), radial $u_y$ (centre column), azimuthal $u_x$ (right column). The radial positions of the blade root and tip are indicated with dashed lines.

The first velocity values which will be inspected are those for the radial traverse. These were sampled at an axial position 0.3 m downstream of the hub point and over the radial segment 0.3 m $\leq r \leq$ 2.68 m. This represents the very near-wake region of the experimental turbine and it should therefore be expected that the filament model works well provided the assumptions of the NLL model are satisfied. To enable comparison with the experimentally collected data, the numerical values have been

averaged over one-third of a rotation. The collected values of the velocity components ($u_x$, $u_y$, $u_z$) and operational points ($U_\infty$: 10, 15 and 24 ms$^{-1}$) are displayed in Fig. 17. In all cases the root, blade span and the external flow regions are clearly discernible. The action of the rotor and the wake is to decelerate the flow in the axial direction and introduce radial and azimuthal velocity components into the flow. The axial velocity $u_x$ represents the principle component from which energy is extracted from the freestream and is therefore a direct metric of the efficacy of the rotor. Discrepancies between experimental

measurements and the numerical predictions of $u_x$ are seen in the root region that grow with increasing inflow velocity. In this radial position, the blade is comprised of the cylindrical root joint and the DU-91 profile, both of which exceed the stall angle at this operating point as seen in Fig. 16 (a). Rather than an ordered vortical flow, the breakdown of the flow in these regions





leads to increased turbulent blockage. In such regions, the weaknesses of the NLL model are directly observable. The shear layer in the tip region (2.0m< $r$ <2.5m) is also observed in all simulations. It can be seen here that increasing the resolution of the VPM method improves the agreement with the experimental shear layer profile. Experimental agreement of the VPM method with a resolution of D/40 is generally better than the FIL method, which implies that for an equivalent grid resolution, the VPM method better predicts the wake velocity profile.

The radial velocity component $u_y$ physically represents the expansion of the streamtube in the near wake. Despite good agreement for the 10 ms$^{-1}$ case, an offset occurs for the 15 ms$^{-1}$ case, which becomes even larger for the 24 ms$^{-1}$ case. In order to find an explanation for this offset, a hypothesis regarding the blade aerodynamics is suggested. This offset may be explained with the assumption that the flow at the blade mid-section (RISØ) is lightly stalled and is therefore exhibiting flow separation. This factor, together with the separated region in the root, would lead to an increase in the turbulent blockage of the rotor and therewith greater streamtube expansion. This may explain both the increase in radial velocity observed in the experiment as well as the overprediction of loads at the blade mid-span for this case as seen in Fig. 15 (e). This argument appears to hold well for the 24 ms$^{-1}$ case as this offset amplifies when the flow strongly separates at the root and at the blade mid-span (RISØ section- see Fig. 16 (a)). The azimuthal velocity component $u_z$ physically represents the streamtube rotation of the near wake. The vortex system generated by the rotor blade induces a wake rotation opposite to the direction to the motion of the rotor, as is seen in experimental data as well as all numerical models. The agreement with the experiment of the VPM model appears to improve with increasing mesh refinement. The agreement of the most resolved model (VPM- D/80) appears to agree well with experimental values up to the radial position where the airfoil separates (see Fig. 16 (a)), suggesting again that the inability of the model to predict separated flow leads to discrepancies with the experimental measurements. Inboard of these regions the flow field encounters an internal shear layer which has no azimuthal component, analogous to the external shear layer in the tip region. In the following the axial traverse velocity measurements made in the experiment are inspected together with the numerical predictions. The axial traverse consists of a set of points where the velocity components upstream and downstream of the rotor were sampled along a the line -4.5 m $\leq$ x $\leq$ 5.9 m, z=0, and at the radial positions r=y=1.5 m. These positions therefore represent a distance of approximately two rotor diameters downstream and therewith the beginning of the far-wake region. Owing to the increase in computational expense incurred by modelling the vorticity-free upstream region of the rotor in the VPM method, only the region -0.5 m $\leq x \leq$ 5.5 m has been modelled with the VPM method. The deceleration of the axial flow ($u_x$) through the rotor is captured relatively well by all resolutions for the VPM method. In general, the agreement with the experimental measurements improves with mesh resolution. An exception can be seen for the 24 ms$^{-1}$ case, where neither the net deceleration of the flow nor the magnitude of the spatial velocity fluctuations due to the tip vortex are well captured. It can be well reasoned that then the net deceleration in the flow will not be well captured by the numerical models if the experimental blockage due to flow separation on the rotor is strong. Inspection of the radial velocity component $u_y$ for this case appears to confirm this point. The inability of the VPM model to capture the spatial velocity fluctuations may be explained through the diffusive effect of the $M_2$ remeshing routine, which acts to spread the mapped vorticity. This may be investigated through the use of a higher order stencil such as the $M_4'$ routine. Regarding the FIL method, the choice of $\delta$=0.2 for wake filaments may too strongly mollify the velocity field induced by the vortex filaments. This may be tested by reducing





this value, however this will lead to stronger discontinuities in the wake pattern and slower convergence of the wake and inflow fields. A similar scenario was encountered in the work by Ramos-García et al. (2017), whereby the predictions of the spatial

velocity fluctuations were reduced as the azimuthal stepsize was decreased. In general, the FIL method appears to have much larger deviations in the predicted velocity than the VPM, again supporting the hypothesis that the VPM method is better suited to wake predictions than the FIL method. These predictions align with similar results presented in Ramos-García et al. (2017).

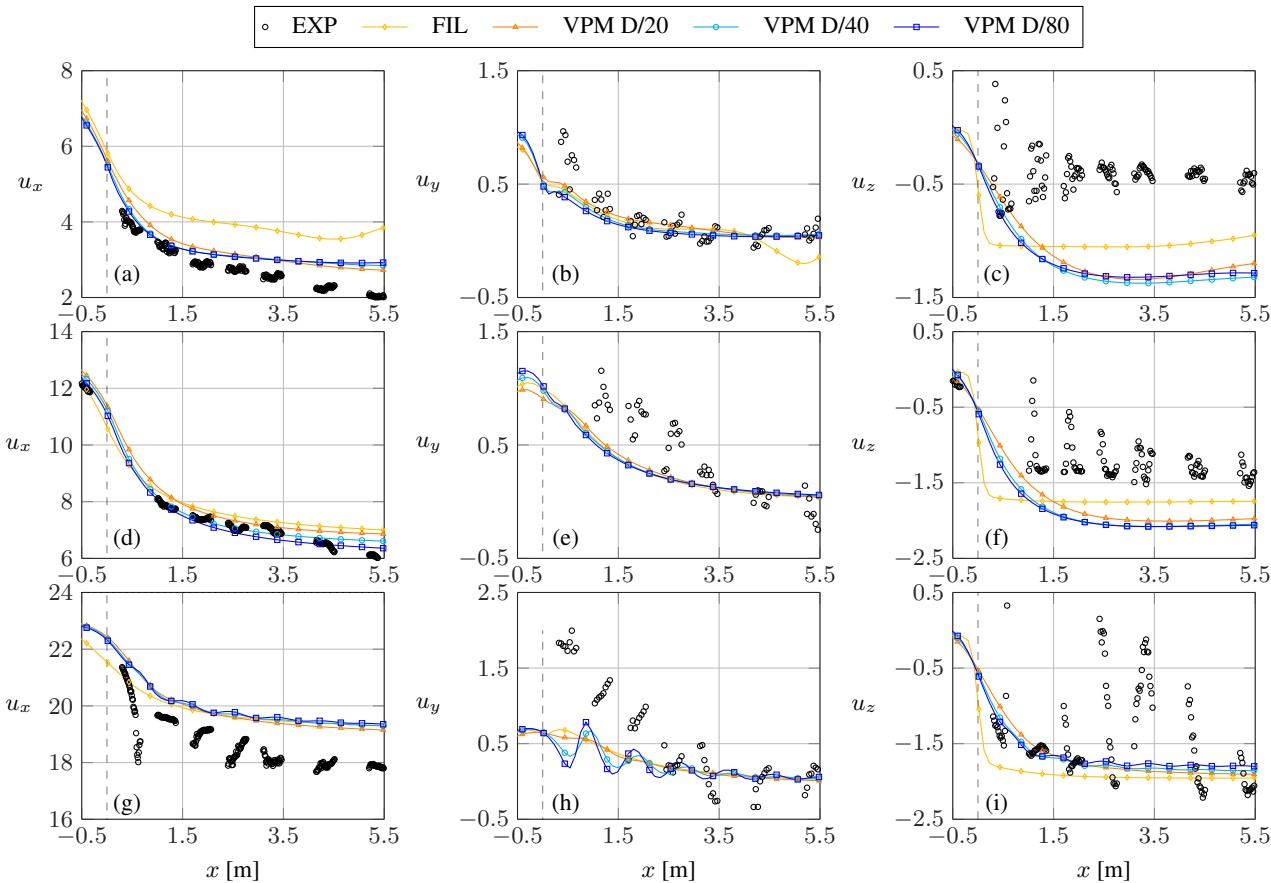

**Figure 18.** Experimental and numerically predicted velocities along an axial traverse at the radial position r=1.5 m. Inflow velocity: 10 ms$^{-1}$ (top row), 15 ms$^{-1}$ (middle row), 24 ms$^{-1}$ (bottom row). Velocity components: Axial velocity $u_x$ (left column), radial $u_y$ (centre column), azimuthal $u_x$ (right column). The axial position of the rotor plane is indicated with the dashed line. For the axial velocity components, the undisturbed inflow velocity is shown (dotted).

Inspecting the radial flow component $u_y$, it is expected physically that streamtube expansion is strongest behind the rotor

and eventually converges to zero as the wake expansion converges, which is directly observable in both the experimental and numerically predicted results. In the 10 ms$^{-1}$ case, the FIL model predicts an increase in the radial velocity component. This case represents a state of streamtube blockage, whereby the induction field is so strong as to force the wake to become unstable





at a much earlier physical position. The physics of this process involve turbulent diffusion and entrainment and are thus far beyond those which can be captured realistically with a potential-flow style filament model. The deviation of the FIL model

here is therefore simply a numerical artefact. A visualisation of the wake predicted by the VPM model for this case can be seen in Fig. 19. For the 15 ms$^{-1}$ and 24 ms$^{-1}$ cases the asymptote of the wake is well captured by both numerical models. The effect of blockage caused by airfoil stall at the root and midspan is again seen to influence the experimental agreement for the 24 ms$^{-1}$ case. The spatial variation in velocity is seen to be predicted by the VPM models and the magnitude of these oscillations improve with increased mesh resolution.


Inspecting the azimuthal velocity component $u_z$, it can be seen that all numerical models predict an approximately constant rotation of the wake after the initial wake expansion. This signifies an approximately rigid rotation of the streamtube, which indeed appears to be observed in the experiments- provided one neglects the spatial variations in velocity caused by the proximity of the tip vortex. For all cases, the VPM model converges with increasing mesh resolution to a value which generally

has a higher magnitude than that observed in the experiment. Observing the experimental values, the wake rotation increases with wind speed. The transfer of momentum to the blades implies an equal and opposite transfer of momentum to the fluid. This explains the increase in wake rotational energy for the higher wind speed due to the greater blade forces - see Fig. 15. This is overpredicted by all models, particularly for the 10 and 15 ms$^{-1}$ cases. Further investigation is required here to determine the root cause of this discrepancy.

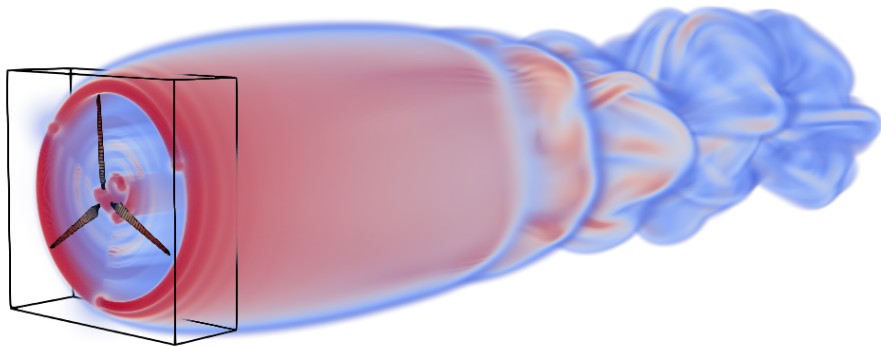

**Figure 19.** The breakdown of the wake predicted due to the high-induction case of the Mexico rotor for and inflow velocity of 10 ms$^{-1}$.

**4  Conclusions**

This paper describes the implementation and validation of a VPM solver as a module of the `SailFFish` library. In the first part of the paper the implementation of the solver is described in detail. The flow field is described by a particle cloud whereby each particle is defined by a position vector, a vorticity vector and a characteristic volume. The particle information is regularly mapped onto an underlying regular grid. The solver's data architecture employs a structure of arrays format for optimised



memory accesses. The resolution of the velocity field is achieved using a fast Poisson solver `SailFFish` which efficiently solves the problem in the frequency domain using fast Fourier transforms. The vorticity field is updated by applying the vorticity transport equation together with an explicit time integration scheme. The required gradients are computed on the regular grid using central finite difference schemes. A particle set reprojection is performed to ensure that the vorticity field remains solenoidal. The novelty of this implementation is the introduction of a discrete region for the solution of the flow over the

lifting body. This allows the flow in this region to be solved with alternative solver types. Within the work carried out here, a lifting line model has been implemented for treatment of lifting bodies.

In the second part of the paper, a number of unsteady flow scenarios were simulated to demonstrate the accuracy and performance of the solver. The first validation was carried out on a flow field without a lifting body; the evolution of a viscous

vortex ring. The diagnostics of the flow field were examined and the expected number of flow field moments were found to be conserved by the solver throughout the translation and dilation of the vortex ring. The evolution of the ring was compared with a number of analytical results available for a viscous vortex rings in low- and high-Re regimes and good agreement was found. An elliptical wing was then simulated. To demonstrate the effectiveness of the lifting line solver an elliptical wing was simulated and the circulation and downwash distributions were found to agree well with steady state analytical results. The

generation and mapping of the unsteady wake vorticity signature and evolution were compared with analytical results for the far wake of an elliptical wing and good agreement was again found.

The accuracy of the solver for helical wake fields was then investigated by comparing the simulation results with the analytical solution of the inflow of an idealised Betz rotor. The influence of the choice of vorticity field regularisations in the

fast Poisson solver was investigated and it was found that increasing the order of the vorticity field regularisation improves the agreement with the analytical result, which was found to converge for increasing resolution and increasing regularisation order. It was found that the choice of a uniform grid can locally lead to errors due to insufficient resolution of the flow field in regions where gradients are strong, such as the tip vortex of a wing. In the final validation case, the rotor of the Mexnext experiment was simulated at three wind speeds. In each simulation, a domain was chosen large enough to allow the wake

flow to converge. Numerous simulated quantities were compared with experimentally measured values. Comparison was also made to a lower-order vortex filament wake model. The steady state loads on the rotor appeared to be in good agreement with the experimentally measured values, however deviations at the medium and high speed cases were observed. These appear to be caused by errors in the provided airfoil polars for the rotor, as was observed in comparable studies in the literature. The near-wake velocity components were compared with results measured on a radial hot-wire traverse directly behind the rotor.

Examination of the axial, radial and tangential velocities showed that numerous flow phenomena were well captured both qualitatively and quantitatively by the model, including the effects of tip and root vortices, wake rotation and expansion. The wake expansion deviated from the experimental values for the medium and high wind speed cases. A suggested hypothesis here is that the flow blockage was not well captured due to incorrect treatment of blade detachment for these operating points. The velocity field along an axial hot-wire traverse extending to approximately 1D downstream of the rotor was also analysed



and displayed comparable results to the radial traverse, wiht th exception that the azimuthal velocity component was slightly overpredicted by all models. In general, the results showed that the velocity field was better captured by the VPM model than the lower-order filament method for an equivalent spatial resolution.

With the validation of the model complete, the VPM solver can now be applied to a number of problems relevant to wind
turbine aerodynamics that require further investigation. These problems may be of a more fundamental nature, such as the stability modes of the wake of a wind turbine operating under idealised (uniform) inflow conditions - as was done in this work. In addition, the influence of inflow turbulence on wake stability modes needs to be investigated to quantify its effect on wake breakdown and recovery. This motivates the next step in the development of the VPM solver, the consideration of non-uniform and turbulent inflow fields. Alternatively, more practical cases can be investigated, including the effect of introducing
disturbances into the wind turbine wake to accelerate wake breakdown and recovery. A number of excitation devices and control methods have been proposed in the literature and it remains to be verified which of these can be practically applied to a full-scale turbine. Application to a range of different scenarios motivates the improvement of the solver performance. The relatively simple block-grid architecture of the solver suggests that the problem is highly amenable to treatment, for example, entirely on a GPU or GPU cluster.

*Code availability.* The fast Poisson solver used in this work is open-source and available online on Github at `SailFFish`. The VPM solver implemented in this work will be made available as a module of `SailFFish`in the near future.

*Competing interests.* The authors have no competing interests to declare.

*Financial support.* This work has received funding by the European Union as part of the Horizon Europe research and innovation project Floatfarm- Grant agreement no. 101136091.



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
