# Peer review of "A Lightweight Vortex Particle-Mesh Library for Variable-Fidelity Simulation of Wind Turbine Wakes"

_Wind Energy Science, 2024_

## Author Comment (AC1)

**Response to RC1**: 'Comment on wes-2024-73'

**General Comment:** This is a response to the extensive review carried out by the anonymous referee #1 on 28th October 2024. It should be noted that I have no intention of continuing with the submission of this manuscript through Wind Energy Science and the paper will be withdrawn. It was my intention to wait for a second review before proceeding, however I believe that the nature of the first review, combined with the difficulties in finding a second reviewer simply warrants retraction of the manuscript.

The purpose for writing this response, despite the intention to retract, is first and foremost to thank the reviewer for the very extensive review. The depth of analysis implies that the reviewer devoted a significant amount of time to reading the manuscript in detail. The second purpose is to provide the reviewer with a response in situations where I wish to rebut, as this too is a crucial part of the interactive peer-review process. I consider the work to have merit for investigations of wake phenomena in wind energy and therefore will continue development and publication in the hope that the review understands my appreciation for his or her input.

I shall proceed in the following by responding in-text to the comments made. In positions where no direct response is made, I understand the reviewer's argument and will take this into account when the manuscript is re-worked.

Joseph Saverin

This paper introduces a Vortex Particle-Mesh (VPM) solver complemented with the handling of lifting bodies, such as lifting lines. This solver is based on the open-source SailFFish library of FFT-based Poisson solvers.

The paper brushes the implementation of such a VPM method and presents comparisons of the code outputs to reference cases and experimental results. The reviewer definitely appreciates the effort made by the author to give a good publicity to the method on methodological aspects and with somewhat extensive validation. However, the implementation itself is essentially identical to previous efforts by groups in DTU (J H Walther), TU Delft (Prof. C S Ferreira), Université catholique de Louvain (G. Winckelmans, P. Chatelain), ETH Zurich (P. Koumoutsakos), Keio University (S. Obi), Université de Pau (P. Poncet, J.-M. Etancelin), CNAM Paris (C. Mimeau), Université de Grenoble (G-H Cottet), Université de Normandie (G. Pinon), etc. These groups have built upon and contributed to about 40 years of research toward turning vortex methods into robust, efficient and mainstream computational tools for fundamental and applied flow problems. The somewhat educated reader cannot help but have these research groups' efforts in mind upon reading the present paper.

- Indeed I am well aware of the work of these groups. Certainly the methodology employed is very similar to other groups, however in my opinion it cannot be stated that the implementation is essentially identical. One core feature of the solver is the fast Poisson solver. A simple investigation of the implementation will show that this is implemented differently than any other codes (of which I am aware). To my knowledge, there are no open-source or openly available vortex particle-mesh solvers developed in c++ and hence any implementation is my own work. I have no other recourse than to directly state that the code is not copied from any other existing software.

This is where this reviewer's main criticism lies. While the author cites a great number of references to the vortex methods literature, this profusion mostly does not translate into the adequate credit in the text (see specific comments section).

- It is unfortunate that the reviewer feels this way. In preparing the manuscript, special effort was devoted to avoiding this- including consulting external parties on this point. I feel that the developments required for the vortex-particle mesh method does not warrant giving credit to every researcher who has ever worked in the field of vortex particle methods, however in places where the reviewer has made specific topics I will keep this in mind.

Moreover, the writing style is often unclear and ambiguous about the methodological originality and novelty: once past the introduction, there is an implicit appropriation of several methodological elements, through e.g. the phrasal modes and tenses used; it can be quite misleading for the reader. This reviewer understands that, in order to present to communicate about an implementation effort, one needs to first present the methodological foundations but one needs to adopt an adequate writing style. As a good example of a review text, I would refer the author to the recent paper by Mimeau and Mortazavi, 2021, that covers the evolution of vortex methods and their recent advances. The present paper lacks an appropriate way for the reader to clearly understand what is new and what belongs to the existing state of the art.

- I appreciate this input from the reviewer and will take it into account when revising the manuscript for submission in a more appropriate journal.

The present work does have some value, mainly on the implementation side and its verification: it presents a (soon-to-be) open-source vortex method implementation. Now, whether that warrants a full Wind Energy Science article, is the true question. Perhaps this contribution would be a better fit for another journal?

- Indeed this is the main motivation for the withdrawal of the manuscript. I personally was of the opinion that implementation papers were a suitable topic for the Wind Energy Science journal, particularly as I myself have read (and reviewed) papers which, in essence, describe implementations. These were ultimately accepted and appear now as journal articles. I appreciate however that the reviewer feels this and in communication with Wind Energy Science perhaps we can work together to ensure that this is emphasized for future authors.

A second contentious aspect is several instances of unsubstantiated claims: e.g., mapping procedures optimizations, etc. (see the specific comments section).

A final global comment is the lack of a coherent motivation and positioning for the proposed contribution. The introduction in its current form does not really conduce to the original part of the work (a lightweight vortex method implementation). What was missing in the current state-of-the-art? What problem does the authors address with this lightweight implementation?

- There were a number of motivations for the contribution. The dominant motivation is to demonstrate (or emphasise) to the wind energy community that the VPM method is a powerful stepping stone between lower order (eg. Filament) method and full conventional CFD. In my opinion, providing a clear demonstration of this type of solver for a number of both analytical and experimental cases accomplishes this. Another motivaiton is the publication of implementation details for a vortex particle-mesh

method (justifying the reviewers' comments on publishing elsewhere). Regarding the current state of the art, I am of the opinion that there is no easily accessible VPM solver available which does not require a significant amount of set up in order to be able to apply it to wind energy simulations.

This reviewer therefore has strong reservations about (1) the contribution put forward, (2) the way it is presented with respect to the existing literature, (3) the quality of the results and of the subsequent discussions, and (4) its true capabilities (see specific comments). In a heavily revised form, the true core contribution of this manuscript would rather be about an implementation and I am afraid that cannot be considered as a topic for Wind Energy Science, given the corpus of methodological works on vortex methods in wind energy and beyond.

**2. Specific comments**

L 59: « higher degree of connectivity can be achieved ... » I disagree, as it is quite the opposite. A particle method loses all connectivity information compared to a filament method, then bringing the issues of keeping the vorticity field divergence-free.

- I understand the argument made and admit that perhaps connectivity purveys the wrong message.

L 68: Chatelain and Koumoutsakos 2010 most certainly did not pioneer so called Vortex-in-Cell methods (which is the subject of that sentence), these methods have been known for two decades before that reference, please check the review of Mimeau and Mortazavi or the works of G-H Cottet.

L 78: the author mentions that the presented solver is conceptually similar to the one of Chatelain et al 2013 but that article also includes the handling of lifting lines. The author does not position, nor motivate their claimed novel approach for the treatment of lifting bodies.

- In my opinion the description of the lifting line model in that reference has only been described in a very succinct manner. A number of points should be considered when implementing such a method and I have attempted to describe these to a reader and justify the chosen "auxiliary grid approach." A large driving force for implementing such an auxiliary approach is that this simplifies coupling the wake solver to other, existing, rotor flow solvers. Evidently this was not communication sufficiently for the reviewer and will be taken into account (this point is emphasized and expounded upon later).

L 91: Did the author use anisotropic mesh spacings? If that mesh is the one used to receive the particle information and reset the particles, then the author has forgotten that the interpolation should also account for the fact that the underlying lagrangian element is anisotropic... and the interpolation back to the mesh is far from trivial... If they just used such a mesh for their simulation without paying attention to this, I would expect some serious numerical issues.

- Isotropic mesh spacings have been used for all implementations. I appreciate the reviewer informing me of these issues and they shall be taken into account shall the code be adapted for anisotropic spacings.

L 119: the storage of the interpolation weights can seem like a good idea until one performs actual timings. Indeed, the interpolation kernels are so cheap to evaluate (polynomials) that the computing gains hardly compensate the additional memory fetching. I would be curious to known whether the authors achieved any gain?

- Gains indeed were seen in the implementation, however the reviewer should keep in mind that the implementation of the VPM solver, as described in this publication, was configured not for clusters or distributed memory cases. Memory fetching therefore has minimal overhead. I would assume that for a distributed memory case, where fetching incurs a larger penalty, the reviewer would be correct. Indeed, in subsequent implementations for GPU devices, where memory transfer between host and device can be the driving factor in performance, the approach suggested by the reviewer was adopted.

L.136: James 1977's contribution is not correctly described. It combines a homogeneous Dirichlet solver on the bounded domain with a second step, unbounded this time, which radiates the sources given by the normal derivate jumps along the boundary (or rather their opposite). It is rather elegant, there is a similar approach by Lackner 1976, thence the references to the so-called James-Lackner algorithm in parts of the literature.

- I have in previous works implemented a James-Lackner method and agree that it is quite elegant. I am therefore aware of the steps in the solver and agree with the reviewer that the phrasing "mixed Dirichlet" is not precise.

L 140: the authors might want to choose their word carefully, as one might understand consistency not only in terms of order but rather a coherence between the FD stencil and the effective FD stencil at work underneath the convolution kernel.. To this reviewer's knowledge, it is the domain of Lattice Green's functions.

L.145: what is the « symmetric nature of the node alignment »? The author also refers to a stencil offset precomputation in the context of some strategy for the FD evaluation on the mesh but no information is given beforehand. It does seem to concern the core of the contribution though (implementation).

- What is precisely meant here is that regardless of the position on the mesh (excluding boundary nodes), the delta of the node id for the neighbors used for FD is identical. This allows for the definition of a template list of relative array positions. This certainly does not represent a significant contribution nor contribute performance and I would hence remove this comment in future versions of the manuscript.

Section 2.4.2 : « magnitude filtering »: rephrase, it is a clipping operation, not a filtering one.

- I was of the opinion that I had seen this phrase be used elsewhere in reference to these methods. The reviewer however is correct either way.

L 183: « novel method proposed by Cottet and Poncet », it is not novel anymore in 2024, and I suspect that there were reprojections in works that predate Cottet and Poncet 2003...

- I used this reference as I had seen it used to reference this method in other publications, however I agree that the approach is probably older and had probably been applied in an earlier work. For subsequent versions I will try to find a more suitable reference.

Sec. 2.5 Treatment of Lifting Bodies: this section relies on an embedded grid, this technique is a long standing challenge for vortex methods, there have been some recent works on the topic, with weak couplings by Billuart et al, J Comp Phys 2022 (contains an assessment of accuracy), and Pasolari et al, 2024.

- Thank you for the relevant reference

In the present work, it is not clear what this embedded mesh achieves compared to other immersed lifting line techniques, such as the one by Caprace et al., 2020. Why the additional mesh? To this reviewer, all could be done within the main computational mesh.

- As a general comment, I believe the embedded mesh is a more general approach as it creates a numerical discretion between the rotor flow region and the wake flow region, for which different solver types may be applied. However, to specifically answer the reviewer's question:

  At each time step, the total velocity field must be evaluated at the blade nodes to solve for the blade circulation. This could be handled, as the reviewer suggests, with the immersed lifting line approach whereby all velocity components are calculated by convoluting the vortex particle field (plus inflow field, blade relative motion etc). If one however wishes to employ a filament-style treatment of the velocity induced by the blade and near-wake, then the flow field is necessarily composed of two regions: The filament region and the particle region. The blade and near-wake (filaments) however naturally have an influence on the particle wake region (particularly in the near field). In the code developed by DTU (MIRAS - DOI: 10.1002/we.2225), this is accounted for by calculating the influence of the filaments on a region of the particle grid using the Biot-Savart law with an appropriate vortex model.

  In the code developed here, an alternative approach is applied. To ensure continuity of the vorticity field on the full domain, at each time step I convert the blade & near wake filaments to their corresponding particle representation. The (full domain) velocity field is then solved using a convolution. This implies that the velocity field extracted along the blade sections includes the effect of the blades and near-wake, which we wish to treat with a filament method. This contribution therefore somehow be subtracted from the velocity field which is sampled for the blade-node velocities. One could carry out a convolution for the velocity field on the full domain accounting for the blade and near-wake vorticity field. This would however be quite inefficient. The embedded grid is therefore used to calculate this velocity component, as this has fewer cells and tight support of the vorticity field of the blades and near-wake. A convolution carried out in this region therefore incurs much lower computational expense.

Note that again, the phrasing is peculiar for works that predate the present manuscript: "similar approach that was taken in Spyropoulos 2022"?!

L 213: "advect the wake nodes..." : Add a reference for this as it is not obvious. Do you use the version with the kink in the trailing vorticity (as done by Phillips, Hunsaker, etc.) or without it (as is Katz and Plotkin)?

- A reference shall be added at this position. The position of the shed wake nodes are calculated based upon three parameters: i) the current TE node position, ii) the "advected" position of the TE node from the previous timestep (a "tracer" particle is emitted there and allowed to advect according to Eq. 4), and iii) a factor which specifies the new filament position based on an interpolation between these two points. I assume by "Phillips, Hunsaker etc" the reviewer means that the wake sheet is advected such that it is not parallel to the trailing edge. With the formulation described above, the angle of

the wake sheet is independent of the trailing edge angle of the blade, and as such the “Phillips, Hunsaker” description better fits the implemented model.

L 216: “an equivalent particle…”: filaments are singular, particles are regular. What happens if a flow particle gets too close to a filament?

- In this solver this does not occur. For the purposes of evolution of the particle set on the “full domain” grid, the lifting body and near wake are always represented by their “equivalent” particle representations. This is described in the response above.

L 220: “ the omega_phi on the embedded grid”: How about the filaments that you did not convert to particles already? And the bound vorticity? These elements dont have a representation on the embedded grid. If their influence is neglected, the solution will be wrong if some extraneous vorticies gets advected in the region of the embedded domain.

- These elements in fact do have a representation on the embedded grid. I have attempted to describe this in L220, however hopefully the text above clarified this somewhat.

L 222: “different velocity field regularisations” : clarify

- Different convolutions between the vorticity and velocity field. For filaments is the filament-type Biot Savart with a van-Garrel type smoothing. For the particles, it is any one of the available regulations available (G2, G4 etc).

L 233: “a line or surface average is found by sampling points” : needs references for this (e.g., melani2024, churchfield2017)

L 237: “Churchfield et al 2017”: they only do 3D mollification; “Schollenberger et al., 2020”: they do 2D, but a better reference for this would be Jha & Schmitz 2018

L 238: “Caprace et al, 2018”: I would argue that the versions of the 1D, 2D, and 3D kernels are due to Caprace 2018 and are not specifically written as a "VPM equivalent", but are rather generally applicable to any type of ALM.

- I agree and will correct this statement.

L 288: “translation velocity”: how do you compute from an instantaneous velocity field?

- The reviewer makes a good point that the method of calculation appears to not be stated. A simple backward FD calculation was applied to the vorticity centroid as defined in Eq (6).

Fig 6 b: the loss of momentum is substantial. It should be conserved by the remeshing procedure (using the Kernel M_2). What is going on?

- In subsequent simulations this appeared to not be an issue. More investigation is needed however I suspect there was an issue with the integration scheme. I have checked this again for this review and indeed with an M4 scheme and double precision, this does not occur and momentum is conserved.

L 308: Eq 9, form for the kinetic energy is not trivial to obtain, please provide reference.

L 328: “validate the solver”: talking about validation is delicate here since there is no real life problem where the solution is the singular lifting line.

- Agreed, this term should be modified to verification.

L 354: “c0 = 1 m and span S = 5 m.” then your aspect ratio is 1.6?

L 357: “h under-relaxation factor of 0.2”: you mentioned an iterative procedure but this was never really described.

- The implementation follows closely that described in a [van Garrel, 2003], where the convergence of the fixed-point iteration is damped by an under-relaxation factor. The section where the model is described will be modified to include this.

Additionally, why do you need underrelaxation here? If all the bound vortex elements are on a straight line, the cross-influence of bound vortex elements on each other is 0 so there is no need to iterate.

- In the lifting line model applied, chordwise filaments are resolved. These therefore have a cross-influence.

Unless you only have 1 straight filament between the bound vortex and 100 wingspans...? but why would you do that? cause it should give you the same solution as the analytical solution.

- I don’t understand the reviewer’s comment here, but I am hoping the above clarifications elucidate the method applied.

L 359: “the trigonometric series solution…”:  There is no difference between this and the actual analytical solution, except that Gamma_ts is a sampled version… but there is no added value in showing this on the plot, it is the same information as Gamma_an, and generates unnecessary noise for the reader.

- I understand the reviewer’s point. The reason for including it was consistency with the following approach where the trigonometric series solution is applied to the rectangular wing. I will rework this section (if it is kept) to clarify the points here.

L 370: “effect of the discretisation”: what happens when one changes the number of elements on the line? Saying that differences are indiscernible is not very committing: rather, can you say what is the minimum number of elements you need to guarantee a given tolerance on the precision?

- This is a good point and this comment should be reinforced.

Also, what will happen when one goes from a cosine distribution to a uniform particle distribution? To be close to the analytical solution, one needs tons of spanwise elements towards the tip. But if you agglomerate this on your grid of the embedded domain, the resulting particle will lose this refinement and the solution suddenly becomes very poor. This is ok if it happens far enough from the bound vortex, but the author never characterized how far. The author only said that the choice of the trigger between filament and particle happens after an arbitrarily chosen number of time steps. What if the time step becomes very small?

- The use of a cosine distribution on the blade indeed means that information is lost if it is mapped to a lower resolution grid. An additional parameter study should be carried out to characterize the effect of the choice of conversion age. I have carried this out in my private investigations, and the results tend to demonstrate that the result conversion both in the limit dT_conv -> 0, and in the limit dT-conv->inf. I have generally chosen the number of elements on the blade to correspond to the number of cells per blade radius, but a more clear comparison would be with a linear distribution and an enforced circulation distribution.

L 376: “This poses numerical challenges for the NLL model,..”: This is unclear.

- The lifting line model of the blade relies upon convergence of the fixed-point iteration applied, which relies on the effect of the induced velocity of the blade elements (and wake) over the lifting body. The stronger the gradient of circulation at the tip, the more unstable this convergence becomes. The case of an elliptic wing is numerically convenient as the element area also decays towards the tip. In the case of the rectangular wing, particle of a low aspect ratio, these instabilities appear to occur and inhibit convergence of the circulation distribution.

The author means that there is difficulty converging your system by only using underrelaxation? well maybe one could do it in pseudo-time an introduce a discretization of the trailing elements in the streamwise direction (akin to a VLM)…

- Yes, this would likely help the convergence.

L 383 “ the NLL model indeed produces accurate results”  but the author had to take some precautions as said earlier… how does that limit the applicability of the method when one puts it in the vpm code?

- It limits the applicability of the method in as such as the parameter required for convergence (under relaxation, core smoothing parameter) etc should be kept in mind when carrying out a simulation. It has generally been found that for high aspect ratio blades for simulations of wind turbines, the model has been relatively insensitive to these parameters. Of course, in cases such as the rectangular wings above or to lifting bodies with a low aspect ratio, I would proceed with great caution, however I would also question whether a lifting line model is appropriate.

L 386 “ comparing the velocity field induced by the VPM model to the vortex filament wake system of the standard NLL method.”: why should they be the same? One is discretized with particles in the wake, the other assumes frozen wake…? or I did not understand? In this section, the author does no't compare velocity fields, they compare downwash

- I have assumed that use of a filament method in the near wake, combined with a conversion to vortex particles should produce similar results in terms of the velocity field induced at the blade. Of course, we expect the results to be slightly different as the velocity field is calculated with two methodologies, however generally the two should not differ greatly. I admit however that this section is generally not well formulated and there are probably issues which would motivate removing it completely. It was conceptually seen also as a way of demonstrating that the NLL model was working, but this isn’t the focus of the paper and as the reviewer suggested this whole section could do with a rewrite (or complete removal). Regarding the use of the word “downwash”; this is erroneous, a better phrase would be “velocity induced at the blade section”

L 393 “4th-order Gaussian kernel”: The author should realize that these two operations (interpolation with M2 and the convolution with the $4^{th}$ order Gaussian kernel) imply a smoothing of the solution. And by the way the smoothing length of the Gaussian kernel needs to be mentioned

L 409: “Caprace et al. (2018)”: There is an entire body of literature on the effect of the regularization on tip loads; these are not the only authors to have dealt with this, check e.g. the works of J N Sørensen,…

- I am aware of these publications, however the purpose of this section is not to express that this work is somehow new or independent, but rather to demonstrate that the model is functioning as expected.

L 413 "In the case of the rectangular wing, no analytical solution exists." If my memories of Prandtl's lifting line basic applications are correct, you can actually use a sum of sine for this. Sure, it is technically infinite, but you can truncate for a given precision. If this precision is greater than the discrepancy with your result, then we can still say something.

- I was unaware that an analytical solution exists to this case. One can adapt the ratio Cl*c of a rectangular wing such that an elliptic lift distribution is attained, however this isn't what is sought. A numerical solution is easily found using the approach of Eq. 11, which is a sum of sines and is therefore, at least numerically equivalent such a solution. I would be very interested in seeing the analytical solution.

L 420 "converges towards accurate results for the ...": No, for this reviewer, the results are not conclusive as no quantitative analysis is performed and the plots do not support that assertion. Plus, the author does not look at what happens for longer vs shorter lengths of the wake that you resolve with filaments. Additionally, they never motivate the use of filaments vs the "Act Line Method"/"Immersed Lifting Line". Why would we want to use this the NLL? To me, it sounds like a step backwards (see general comment about section 2.5 above)

- I understand the reviewer's viewpoint here as to the approach being a step backwards rather than a step forwards. As mentioned above I believe this approach can be attractive for certain applications. The chosen application case here was envisioned as coupling onto another software package which has a filament treatment, ie- a intermediate approach was desired. In addition to this, the results generated by Ramos-Garcia et al. (numerous publications listed in references) indicate that the combined filament-particle approach produces desirable results. The code, however, has been designed such that the implementation of an ACL model would not require any structural changes to the code and is in fact planned as future work.

3.2.4 Unsteady Case- Wake Rollup: This section would benefit from a thorough rewrite.

The terms used are confusing: the vortex pair has to converge to a separation between centroids of pi b/4, simply by conservation of momentum. Please clarify displacement, separation etc.

How are the C_y and epsilon_inf quantities measured within the simulation?

- On L447 I mentioned that the vorticity centroid has been used and that the average velocity has been used to calculate eps_inf. These are calculated over a plane downstream of the rotor.

L 430 : if the author has to invoke computational costs and bypass the transient for such a simple problem, there is a problem with the whole point of the paper: an efficient lightweight vortex particle method.

- Agreed, I didn't need to avoid an unsteady simulation. This is a relatively simple case also with unsteady wake development. The real motivation is not simulation time, but rather ensuring that the circulation is precisely as defined in Kaden, and circumventing any possible deviation from this caused by the use of the NLL model.

L 433: "carried out with M_2 scheme." This is again very diffusive, so while the author has no viscosity, the effective viscosity due to this interpolation kernel will overwhelm the simulation

- Agreed, if this section is in fact kept in the persisting manuscript, it makes much more sense to use an M4 or M6 interpolation scheme.

L 448: "the average velocity field in z direction is sampled to calculate the wake downwash": this is related to a question for sec 3.2.4, but it is still not clear: how, with what smoothing for the sampling?

- No smoothing was applied, simply an average.

L 451: "The metric Cy is deceptive as the relative motion of the tip vortex is…": this comment comes after you have used this quantity to claim convergence on the previous page!?

- Now, with retrospect, I don't understand why I wrote this. The motion is not small when compared to the dimension of the wing.

L 454 "Furthermore,…": this is not convincing. Examples of accurate vortex-in-cell, or particle-in-cell simulations abound, the author should consider reviewing the 30 or so years of literature on these methods.

Sec 3.3.3 Unsteady Case: Mexnext Rotor:

L 554: the discussion of the lifting line model limitations is not entirely correct. An airfoil in a separated flow regime (stall) can be handled by a lifting line, at a time sufficiently large to be able to identify the attached circulation as the one comprising the airfoil bound circulation and the one within the recirculations. The Kutta-Joukowski theorem still holds for that total circulation. The limitations then rather lie in the three-dimensionality of the flow and spanwise gradients (and by extension tip effects…).

L558: "Fig. 15 (b) is very similar to that predicted by similar NLL models in the works of Boorsma": none of these results are shown. The discussion is impossible to follow/

- If the reviewer feels that this is crucial, then this should be added in the corrected manuscript. More input to the difficulty of the discussion would be helpful. The paper had been internally reviewed by another person and such an impossibility was not flagged, however I appreciate that the general text could be made to flow better.

L 600 : The discussion of the aerodynamic loads should be revised and properly structured. Furthermore, the conclusions mention an agreement (trends and magnitude) with the experiments; this is quite an exaggeration.

- I will carry out a rewrite of the section on loads and attempt to improve readability. In the majority of cases I can see agreement (trends and magnitude) and if I compare with other comparable results (e.g. Ramos-Garcia 2017), the agreement is as good if not better and was used in that publication to justify a successful validation of the solver.

L 602-679: The agreement was only on trends for the loads; how can the author expect any agreement with the experiments on the wake velocities?

- The comment itself is correct, incorrect loads will naturally give rise to errors with the wake velocity predictions. This comment I feel is relevant for the 24 m/s case, however for the 10 and 15 m/s case agreement of magnitude of blade loads was also observable.

Figure 19: At last, this figure confirms the overwhelming role of the interpolation kernel used. Individual vortices disappear within a fraction of a revolution, due to the resolution and the M2 interpolation. Additionally, one can identify shed individual vortices at mid-span/radius along the blade, one would expect to observe steps/gradients in the circulation (lift) at the locations of these vortices in Figures 15 and 16.

- The reviewer has a valid point here. The use of an M2 kernel will greatly influence results and I will carry out the analysis again using an M4 or M6 -type interpolation to investigate the effect this has. The resolution of the grid shown here is also not particularly high, which gives rise to an increase in the smoothing effect of the mentioned interpolation. In subsequent reviews of the results and solver I have found an additional problem with the interpolation scheme which may have contributed to an over-smoothing of the solution field. The results will be re-examined prior to any resubmission.

Conclusions:

L 681: the author has performed mainly verification; the experimental results do not validate the models used (the airfoil polars to be fair)

L 694: conclusions about vortex ring case: this case was not quite challenging and the Reynolds numbers were not high. Challenging cases are those of ring collisions, or flows like a Taylor-Green vortex flow

- This is true, however a serious demonstration of these cases would also necessitate a lot more "space" in a publication for a wind energy journal. Ideally the comment of the reviewer is accurate that such a validation (for this section at least) perhaps belongs better in another journal.

L 724-735: Some of the final comments are a bit over-optimistic

- Validation, as mentioned above, should rather be verification
- The capture and prediction of wake alleviation schemes will require finer resolutions and a high order interpolation kernel. Such problems will also involve unsteady vortex shedding, something that has not been tested here and that could prove challenging for the NLL – VPM coupling. To be honest, the NLL steady flow assumption will probably greatly affect the results.
    - Agreed on the first two points. I assume that with vortex shedding the reviewer means from the tower, nacelle or blade roots? Regarding the comment the NLL steady flow assumptions, I am unsure what is meant here. To the application of vortex shedding? This I would agree with.
- The port to GPU for particle-grid schemes is not so simple due to the relatively low computational intensity per degree-of-freedom. A simple datastructure does not automatically translate into large speedups when making the jump to GPU.
    - Preliminary testing has demonstrated that this indeed is true, however significant speed-ups of the order of approx. factor 30 have been found for a shared memory implementation. For distributed memory applications, simple porting to GPU would be even more challenging and probably be less advantageous.

- The manuscript actually contains very little information about the implementation itself, which is the one original contribution in this work. Some aspects are touched in several places in the manuscript, assuming the software architecture and the data structures have been presented. It is not the case and make the read quite difficult.
- Finally, the VPM client, announced as open-source, is found not to be available yet.
    - The VPM client has not been announced as open-source. The only comment referring to open-source is the fast-Poisson solver applied for the vorticity-velocity convolution and that it is planned to release the solver open-source.

**3. Technical corrections**

L 65: Greengard and Roklin, A fast algorithm for particle simulation,... 1997 should read **1987**.

- This would be the case if I was referencing one of the original papers on the FMM, I am however reference a later paper. See doi.

L.135: James 1977, there is no journal, (should be Journal of Computational Physics)

Eq 3: matter of notation : the time derivative is a material one (noted D/Dt typically, and not as a total derivative as would be the case for particle positions and strengths)

L. 181: a divergence-free vorticity is the consequence of vector analysis (div of a curl is null)

L. 218 : “Eulerian grid”: isn’t G_L supposed to be the Lagrangian grid? Clarify

L 269-270: some typos or confusions between t_delta and t_Gamma?

L 285: symbol “H” not the same as the “h_x” etc. introduced earlier, correct or clarify

L 343, eq 11 and the text: Using "S" for the span is very very confusing... and what is b then?

- b = S/2. This is (erroneously) not mentioned in the script. I should proceed simply with b.

L 357: “alpha = 5.7106” why so much precision for an initial value?

- This is the result of using an inflow u =[1.0, 0.0, 0.1]. Of course this should be mentioned, otherwise the value on its own is questionable.

L 398 “the downwash predicted by the filament method is very near the analytical result.” : already shown previously

L 400 “error” : these are two different models, so one should not refer to this as an error but rather a difference or a discrepancy

L 616: “of the cylindrical root joint and the DU-91 profile, both of which exceed the stall angle”: As it is written, the sentence mentions the existence of a stall angle for a cylinder...

- Agreed, this comment is, as written, erroneous.

**4. References**

Chloé Mimeau, Iraj Mortazavi. A Review of Vortex Methods and Their Applications: From Creation to Recent Advances. Fluids, 2021, 6 (2), pp.68. ⟨10.3390/fluids6020068⟩.